# Observations of Atmospheric Chemical Deposition to High Arctic Snow

Katrina M. Macdonald[1], Sangeeta Sharma[2], Desiree Toom[2], Alina Chivulescu[2], Sarah Hanna[3], Allan K. Bertram[3], Andrew Platt[2], Mike Elsasser[2], Lin Huang[2], David Tarasick[4], Nathan Chellman[5], Joseph R. McConnell[5], Heiko Bozem[6], Daniel Kunkel[6], Ying Duan Lei[1], Greg J. Evans[1], Jonathan P. D. Abbatt[7]

[1] Department of Chemical Engineering and Applied Chemistry, University of Toronto, M5S 3E5, Canada
[2] Climate Research Divisions, Environment and Climate Change Canada, Toronto, M3H 5T4, Canada
[3] Department of Chemistry, University of British Columbia, Vancouver, V6T 1Z1, Canada
[4] Air Quality Research Divisions, Environment and Climate Change Canada, Toronto, M3H 5T4, Canada
[5] Desert Research Institute, Reno, 89512, Unites States of America
[6] Institute for Atmospheric Physics, Johannes Gutenberg University Mainz, 55128, Germany
[7] Department of Chemistry, University of Toronto, M5S 3H6, Canada

*Correspondence to*: Jonathan Abbatt ( jabbatt@chem.utoronto.ca)

**Abstract.** Rapidly rising temperatures and loss of snow and ice cover have demonstrated the unique vulnerability of the high Arctic to climate change. There are major uncertainties in modelling the chemical depositional and scavenging processes of Arctic snow. To that end, fresh snow samples collected on average every four days at Alert, Nunavut, from September 2014 to June 2015 were analyzed for black carbon, major ions, and metals, and their concentrations and fluxes reported. Comparison with simultaneous measurements of atmospheric aerosol mass loadings yields effective deposition velocities that encompass all processes by which the atmospheric species are transferred to the snow. It is inferred from these values that dry deposition is the dominant removal mechanism for several compounds over the winter while wet deposition increased in importance in the fall and spring, possibly due to enhanced scavenging by mixed-phase clouds. Black carbon aerosol was the least efficiently deposited species to the snow.

## 1 Introduction and Background

In recent decades drastic changes have been observed within the Arctic, including a rapid increase in surface temperatures and loss of sea ice and snow cover (Rigor et al., 2000; Stroeve et al., 2005; Hartmann et al., 2013). Not only have these changes had adverse consequences for local populations and ecosystems, it has been suggested that their impacts may be significant at the global scale (Law and Stohl, 2007; AMAP, 2011). Light-absorbing compounds, the most widely-studied of which being black carbon (BC) particles, can have a particularly significant impact on the Arctic atmosphere and snow systems through the absorption of solar radiation and subsequent warming and snow melt (Bond et al., 2013). While the Arctic atmosphere has been previously explored spatially, temporally, and compositionally (e.g., Hartmann et al., 2013), Arctic snow and the mechanisms linking snow to the atmosphere have been the subject of only a relatively small number of

studies (AMAP, 2011), despite the enormous amount of research conducted on the Arctic Haze phenomenon (Quinn et al., 2007). Seasonal observations of fresh snow samples are particularly uncommon (e.g., Davidson et al., 1993; Toom-Sauntry and Barrie, 2002; Hagler et al., 2007) and previous explorations of snow deposition and scavenging mechanisms have been largely reliant on short-term or aged snowpack sampling (e.g., Bergin et al., 1995), ice cores (e.g., Legrand and De Angelis

1995), modelling, and laboratory tests.

Aerosols entering the Arctic atmosphere, either generated locally or transported from elsewhere, can be removed by atmospheric transport or deposition. Deposition of particles follows two mechanisms: dry deposition, whereby particles are deposited to the ground by impaction, gravitational settling, and Brownian motion; and wet deposition, whereby particles are scavenged by hydrometeors and deposited through precipitation. Wet deposition is further split into two scavenging

mechanisms: in-cloud scavenging which removes particles from the cloud layer during precipitation formation, and below-cloud scavenging which removes particles from the atmospheric column through which precipitation falls. Gaseous compounds also undergo similar scavenging processes (Seinfeld and Pandis, 2006).

The rate of dry deposition is dependent on the properties of the depositing particle, the surface onto which deposition occurs, and the air-surface boundary layer (Sehmel, 1980; Zhang and Vet, 2006). Dry deposition velocities of accumulation mode

particles, the dominant mass-weighted mode of particles observed in the non-summer Arctic (Sharma et al., 2013), to snow have been modelled and observed over a range of 0.01 to 0.60 cm/s, typically within 0.02 to 0.10 cm/s; gaseous deposition velocities to a snow surface show a similar range, with observations from 0.05 to 0.50 cm/s and a typical velocity of approximately 0.10 cm/s (McMahon and Denison, 1979; Sehmel, 1980; Davidson et al., 1985b; Davidson et al., 1987; Hillamo et al., 1993; Bergin et al., 1995; Petroff and Zhang, 2010; Liu et al., 2011). Wet deposition is dependent on the

properties of the depositing aerosol and the atmospheric conditions. In-cloud scavenging is largely controlled by a particle's size and composition which dictate its ability to nucleate hydrometeors and to be scavenged by cloud droplets. Particles can act as cloud condensation nuclei (CCN) that nucleate water droplets or ice nuclei (IN) that nucleate ice crystals. Common CCN components include sea salt, sulphate ($SO_4^{2-}$), and nitrate ($NO_3^-$), while mineral dust and bioaerosols are common IN (Hoose and Möhler, 2012; Farmer, Cappa, and Kreidenweis, 2015). Typically, BC is considered to be an ineffective CCN or

IN (Hoose and Möhler, 2012; Farmer, Cappa, and Kreidenweis, 2015). Although liquid water clouds are not expected during the Arctic winter, mixed-phase clouds, which contain both liquid water and ice, have been observed in this region at temperatures well below 0 °C, in unusual cases down to -40 °C but more commonly between -20 °C and -10 °C (Morrison et al., 2005; Shupe et al., 2006). Below-cloud deposition is dependent on snow type and meteorological conditions, which dictate the volume of air scavenged per snowfall (Zhang and Vet, 2006). Particle size also affects below-cloud scavenging

with higher scavenging efficiencies for particles above 2.5 μm diameter relative to accumulation mode particles (Zhang and Vet, 2006). The total wet deposition is a function of the volume of precipitation. Hence, wet deposition is not typically described via a deposition velocity.

The goals of this paper are to present a new dataset in which the chemical composition of freshly fallen snow was measured through a fall-winter-spring period at a high Arctic field station. By combining these data with simultaneous measurements

of ambient aerosol, the efficiency of deposition of individual species from the atmosphere to the snow can be evaluated under a set of broad assumptions. While this paper presents the measurement data set in detail and focuses on the depositional and scavenging mechanisms that can be inferred from it, a subsequent publication will identify potential pollutant sources based on the snow compositional data. To our knowledge, this is the first time that the composition and

flux of freshly fallen snow has been analyzed at high temporal frequency throughout an entire cold season in the high Arctic. All data from this study will be available upon conclusion of the NETCARE project via the Government of Canada Open Data Portal.

## 2 Methodology

### 2.1 Snow Sample Collection

Snow samples were collected at Environment and Climate Change Canada's (ECCC) Dr. Neil Trivett Global Atmosphere Watch Observatory at Alert, Nunavut from September 14[th], 2014 to June 1[st], 2015 as part of the Network on Climate and Aerosols Research (NETCARE) initiative to create a temporally-refined and broadly speciated dataset of high Arctic snow measurements. Alert is a remote outpost in the Canadian high Arctic, at the northern coast of Ellesmere Island (82°27' N, 62°30' W), with a small transient population of research and military personnel (location details provided in supplemental

Sect. S4). Snow samples were collected from two Teflon-surfaced snow tables (about 1 m$^2$ by 1 m above ground level, shown in supplemental Figure S2) located in an open-air minimal traffic site, about 6 km SSW of the Alert base camp, 201 m above sea level. Freshly fallen snow was collected from the tables using a Teflon scraper and scoop by dividing the table into rectangular portions for replicate sample collection. Four replicate samples were collected for this study and the table area cleared to fill each bottle was recorded. Prior to their first use and between snow sample collections, both snow tables

were fully cleared of all remaining snow and cleaned with methanol. Samples were collected as soon after the end of each snowfall as feasible, conditions allowing. From September 14[th], 2014 to June 1[st], 2015, 59 sets of snow samples were collected. When insufficient snow volume was available for complete collection, a subset of the replicate samples was collected as listed in supplemental Table S1. The interval between collections varied based on snowfall frequency, ranging from 1 to 19 days with an average of 4 days. The table area and collection period length associated with each sample allowed

the measured concentration of each analyte to be converted to a flux. New sample bottles were used for the collection campaign and each bottle was thoroughly cleaned prior to use. Bottles and lids were soaked in 5% nitric acid, 1% detergent in water (Alconox), and deionized 18.2 MΩ water (DIW), allowing 10 to 14 hours for each soak. Each dried bottle was then sealed in a protective plastic sleeve until use. At Alert, plastic outer gloves and lab coats were used to minimize contamination during collection, and the scraper and scoop were cleaned with DIW prior to each collection.

The collection of fresh snow samples reduces the impact of snow sublimation and/or melt as well as the movement of chemical species between snow and air, which can be a concern for snowpack sampling; however, some bi-directional exchange between snow and atmosphere is unavoidable within natural snowpack and still expected to smaller extent on the

snow table. Also, the collection of samples from a snow table eliminated the difficulty in distinguishing the fresh stratigraphic snow layer from aged layers below, a source of uncertainty for traditional surface snow sampling. This ability to assign a well-defined deposition area and time period to each sample was an advantage over traditional sampling campaigns of aged snowpack. However, both this and traditional snow collection techniques are prone to the uncertainty

introduced by the redistribution of snow by winds. Measurements of snowfall accumulation were not available for the collection site. Snow depths measured at the Alert ECCC station indicate that the snow collected on the tables may have underestimated the total snowfall volume by a factor of approximately 1 to 10; however, the meteorological station and collection site were separated by over 6 km with a 50 m difference in elevation, and there was significant disagreement between operator records of weather and that indicated by the meteorological station (see supplemental S4.2 for details).

Thus, it was unclear whether this disagreement was the result of snow loss from the snow table or the natural spatial variability in precipitation, and no correction was applied to the collected snow depth. Furthermore, it should be noted that dry deposition via the filtration of air as it is pumped through the snowpack (as described in Harder et al., 1996) may differ between snow on a snow table and that on the surface.

### 2.2 Campaign Meteorological Conditions

Alert station operators recorded the collection conditions for each sample. Atypical snowfall events were noted: diamond dust events, small crystalline snowfalls; and blowing snow events, periods when high winds potentially resuspended snow from the ground. Operators also made note of any unusual weather conditions such as fog or blizzard conditions. Local ground-level meteorological conditions were monitored by the Alert ECCC stations, approximately 6 km NNE of the collection site (station IDs 2400306, 2400305, and 2400302; retrieved Nov. 2015 from climate.weather.gc.ca). In addition to

ground-level meteorological information, vertical profiles were monitored via 6 to 12-hour radiosondes. The radiosonde data were used to estimate mixing height and cloud height over the campaign. Mixing height was taken as the lowest altitude corresponding to an inflection point in the potential temperature. If the potential temperature gradient did not change from negative to positive within the lowest 3 km, no mixing height was found. The vertical humidity profiles were used to identify cloud height as the lowest altitude, within 3 km of the surface, with 100% relative humidity. When 100% humidity was not

reached, this criterion was relaxed to 95%. Details of meteorology data are provided in the supplemental section S4.2.

### 2.3 Snow Sample Preparation and Analysis

All snow samples were kept frozen prior to analysis, throughout storage and shipping. A broad suite of analytes was quantified using replicate snow samples from each snowfall: BC, major ions, and metals. Detailed procedures are provided in the supplemental S2.

Refractory BC quantification was completed via single-particle soot photometry (SP2) as per McConnell et al. (2007). Briefly, melted and sonicated snow samples were atomized via Apex-Q nebulizer and dried particles with 0.02 to 50 fg BC

were quantified via SP2. Observed BC mass distributions did not suggest significant underestimation of the total BC mass due to this size cut-off. A quality control standard and analysis blank were analyzed for every batch of 17 samples.

Major ions were measured via ion chromatography (IC) at ECCC, as per Toom-Sauntry and Barrie (2002). Briefly, melted samples were quantified using a Dionex IC: DX600 for anions and cations, ICS2000 for organic acids. Aliquots of these samples were also used for pH analysis (Denver pH analyzer). Equipment was calibrated daily and quality control runs completed every ten samples.

Metals analysis was completed via inductively coupled plasma mass spectrometry (ICP-MS) at the University of Toronto. Briefly, melted samples were filtered to separate insoluble and soluble metals (considered as that which was retained or passed through a 0.45 µm cellulose acetate filter, respectively). Both filtrate and filter were digested using 70% nitric acid, ultra-trace grade (SCP Science PlasmaPure), and filter digestion was augmented using a microwave digester (CEM MARS 6). Centrifuged samples were then quantified via ICP-MS (Thermo Scientific iCAP Q). A performance test and calibration (SCP Science PlasmaCAL QC Std 4) were completed prior to each run, and quality control checks were completed every ten samples. Also, an internal standard was included to quantify and correct for any instrument drift or inter-sample variability (SCP Science Int. Std. Mix 1). All sample preparation was completed in a class 100 vertical laminar flow cabinet (AirClean Systems AC 632).

Quality assurance is of the upmost importance in the analysis of dilute Arctic samples. Instrument accuracy was confirmed through the analysis of certified reference materials. The uncertainty of each measurement was estimated based on analysis detection limits and reproducibility; details are provided in supplemental Sect. S2 (as per Reff et al., 2007; Norris et al., 2014). Also, the signal-to-noise (S/N) of each analyte was calculated to indicate the strength of each measurement, with a S/N value over one considered to be strong (Norris et al., 2014). Regular analysis of blanks was used for background subtraction and to define method detection limits (MDL) as three standard deviations of the blank levels. Beyond typical preparation blanks, which used DIW in the place of snow melt water, field blanks were also analyzed. Once per month, a set of empty sample bottles was taken to the snow table, opened, and resealed without collection. These field blank bottles were stored and shipped with the regular samples and rinsed with DIW to quantify any contamination throughout the sampling process. Any influence from the local Alert base camp was identified using local wind records and the activity logs of the base camp personnel. The only analytes that showed a potential influence from base camp winds were crustal metals, with Pearson's correlation coefficients (R) of 0.4 to 0.6 (p-value 0.0001 to 0.002) between snow mixing ratios and periods of base camp winds. Base camp combustion activity logs showed no significant impact on the samples.

## 2.4 Atmospheric Monitoring

Ground-level atmospheric monitoring data from the Alert Global Atmospheric Watch Observatory were provided by ECCC (see supplemental S3 for details). Atmospheric BC was monitored hourly by SP2 (Droplet Measurement Technology) (as per Schroder et al., 2015) and major ions by IC of 6 to 8-day high-volume filters of total suspended particles (Hi-Vol) (as per Sirois and Barrie, 1999). Both the SP2 and Hi-Vol were operational throughout the campaign with coverages of 92% and

94%, respectively. As in the snow SP2 analysis, the influence of BC particle size cut-off limits was considered but found to have minimal impact, especially when considered at a monthly scale.

## 2.5 Transport Modelling

The Lagrangian particle dispersion model FLEXPART (Stohl et al., 2005) was used to determine the source region of air masses that were measured over Alert. This model has previously been shown to be an effective tool for the prediction of transport pathways into and within the Arctic (e.g., Paris et al., 2009). The simulations were driven using meteorological analysis data from the European Centre for Medium-Range Weather Forecasts with a horizontal grid spacing of 0.25° in longitude and latitude and 137 levels in the vertical. For each 5 day period during the measurement period we released virtual tracers over Alert in four different altitude levels, 100 m, 500 m, 1000 m, and 2000 m above sea level, to distinguish between the levels which may be scavenged by snow. The tracers were then followed for ten days backward to obtain the source region for the particular time period. The FLEXPART results were used to explore the dominant source regions associated with each sample. As a simple quantification of the variability in source region, FLEXPART trajectories were summarized by the observed southern limit of transport. This southern limit was calculated as the latitude which encircles 98% of the 10-day transport source area. Values were calculated using each of the four initialization altitudes.

## 3 Results and Discussion

### 3.1 Total Deposition of Arctic Snowfall Events

Each sample for this study was collected fresh after a known time and over a known area. Given that the snow tables were exposed to the ambient atmosphere for the entirety of each collection period, the measured deposition is considered to represent the total deposition (wet and dry) for said period; however, it is known that surrogate surfaces do not provide an exact proxy for the deposition which would be seen to a natural snow surface (Ibrahim, Barrie, and Fanaki, 1983; Davidson et al., 1985a; Hicks, 1986). There are two additional caveats to this assumption. Firstly, dry deposition at the beginning of each period would fall directly on the exposed clean table rather than onto previously deposited snow. It is unknown what impact these different surface characteristics could have had on the initial deposition rate and collection efficiency. Thus, there is additional uncertainty in the capture of initial dry deposition to the bare table. Secondly, strong winds can disturb and redistribute the snowpack and cause snow to be blown off and/or onto the snow table. Alert operators recorded four occasions when the snowpack was observed to be resuspended due to high winds and these were excluded from the presented results. The dates of these blowing snow events are noted in Table S1 as are missed collections.

The observed snow mixing ratios and fluxes are summarized in Table 1 and Figure 1 for measured analytes with a strong S/N. Mixing ratio is reported as parts per billion by mass (ppb) with the exception of pH. Flux is reported on a per day basis to take into account the differing collection period lengths; however, it should be noted that this length corresponds to the entire collection period (i.e., the number of days between clearing the snow table), not just the length of time when snow was

actually falling. A full record of the measured deposition over the campaign is provided in the supplemental (Table S1-6) along with the associated uncertainties and notes of atypical collection conditions. It should be noted that although IC measurements are provided as the measured ions throughout the discussion, these analytes may not necessarily exist in the dissociated ionic form in the environment. Also, the metal measurements provided in Table 1 are total values, insoluble and soluble. The soluble fractions differed by analyte and by date and are provided in the supplemental (Table S4-6). The metal measurements can be roughly classified into three categories: predominantly insoluble analytes Fe and Al (>50% insoluble over full campaign); variably soluble/insoluble analytes Co, V, As, Cu, Pb, Mn, K, and Mg; and predominantly soluble analytes Ca and Na (<50% insoluble) (in order from least to greatest average soluble fraction), excluding analytes with insufficient soluble or insoluble measurements above MDL.

**Table 1: Overview of fresh snow composition and inferred fluxes over the 2014-15 winter season.**

| Analysis | Analyte | Snow Mixing Ratio (ppb) | | | Snow Flux ($\mu g/m^2/d$) | | |
|---|---|---|---|---|---|---|---|
| | | $25^{th}$ Percentile | $50^{th}$ Percentile | $75^{th}$ Percentile | $25^{th}$ Percentile | $50^{th}$ Percentile | $75^{th}$ Percentile |
| SP2 | BC | 1.3 | 2.3 | 4.1 | 0.24 | 0.42 | 0.86 |
| IC | MS | <1.9 | <1.9 | 2.5 | <0.1 | <0.1 | <0.1 |
| | ACE | 9.6 | 19.9 | 27.3 | 1.9 | 3.5 | 7.5 |
| | PRP | <1.5 | 2.2 | 5.3 | <0.10 | 0.62 | 2.06 |
| | FOR | 8.4 | 11.0 | 14.8 | 1.32 | 2.66 | 4.72 |
| | $Cl^-$ | 132 | 249 | 605 | 35.6 | 59.2 | 122.5 |
| | $Br^-$ | <5.0 | <5.0 | 12.1 | <0.3 | <0.3 | 2.0 |
| | $NO_3^-$ | 85.4 | 152.5 | 265.8 | 10.8 | 23.8 | 50.2 |
| | $SO_4^{2-}$ | 204 | 297 | 554 | 32.4 | 69.9 | 132.5 |
| | $C_2O_4^{2-}$ | <18.0 | <18.0 | 20.8 | <1.2 | 0.2 | 2.7 |
| | $Na^+$ | 55.4 | 110.7 | 237.9 | 10.9 | 20.1 | 52.4 |
| | $NH_4^+$ | 10.6 | 12.4 | 16.6 | 1.3 | 2.5 | 5.9 |
| | $K^+$ | 8.0 | 15.4 | 23.5 | 0.8 | 2.0 | 3.9 |
| | $Mg^{2+}$ | 22.3 | 43.3 | 77.4 | 2.2 | 7.6 | 13.3 |
| | $Ca^{2+}$ | <133.1 | 193.1 | 409.2 | <9.1 | 14.6 | 51.5 |
| pH Analyzer | $H^+$ (pH) | 1.39 (5.16) | 4.25 (5.37) | 6.97 (5.86) | 0.25 n/a | 0.88 n/a | 1.87 n/a |
| ICP-MS | Mg | 18.2 | 28.6 | 67.6 | 3.0 | 6.5 | 14.5 |
| | Al | <3.2 | 7.2 | 19.2 | <0.2 | 1.4 | 3.3 |
| | V | 0.006 | 0.012 | 0.086 | 0.002 | 0.003 | 0.014 |
| | Mn | 0.23 | 0.64 | 1.14 | 0.06 | 0.10 | 0.20 |
| | Fe | 3.6 | 10.8 | 29.1 | 0.5 | 2.0 | 3.9 |
| | Co | <0.002 | 0.004 | 0.011 | <0.0002 | 0.0007 | 0.0015 |
| | Cu | <0.02 | 0.05 | 0.28 | 0.001 | 0.010 | 0.053 |
| | As | 0.007 | 0.044 | 0.071 | 0.002 | 0.006 | 0.013 |
| | Se | 0.010 | 0.024 | 0.058 | 0.002 | 0.004 | 0.010 |
| | Sb | 0.004 | 0.010 | 0.018 | 0.001 | 0.002 | 0.004 |
| | Tl | <0.0001 | 0.0001 | 0.0004 | $<7.2\times10^{-6}$ | $15.6\times10^{-6}$ | $54.0\times10^{-6}$ |
| | Pb | 0.05 | 0.25 | 0.41 | 0.012 | 0.039 | 0.086 |

Notes:  BC = black carbon; MS = methanesulphonate; ACE = acetate; PRP = propionate; FOR = formate.
< # indicates measurement is below MDL.

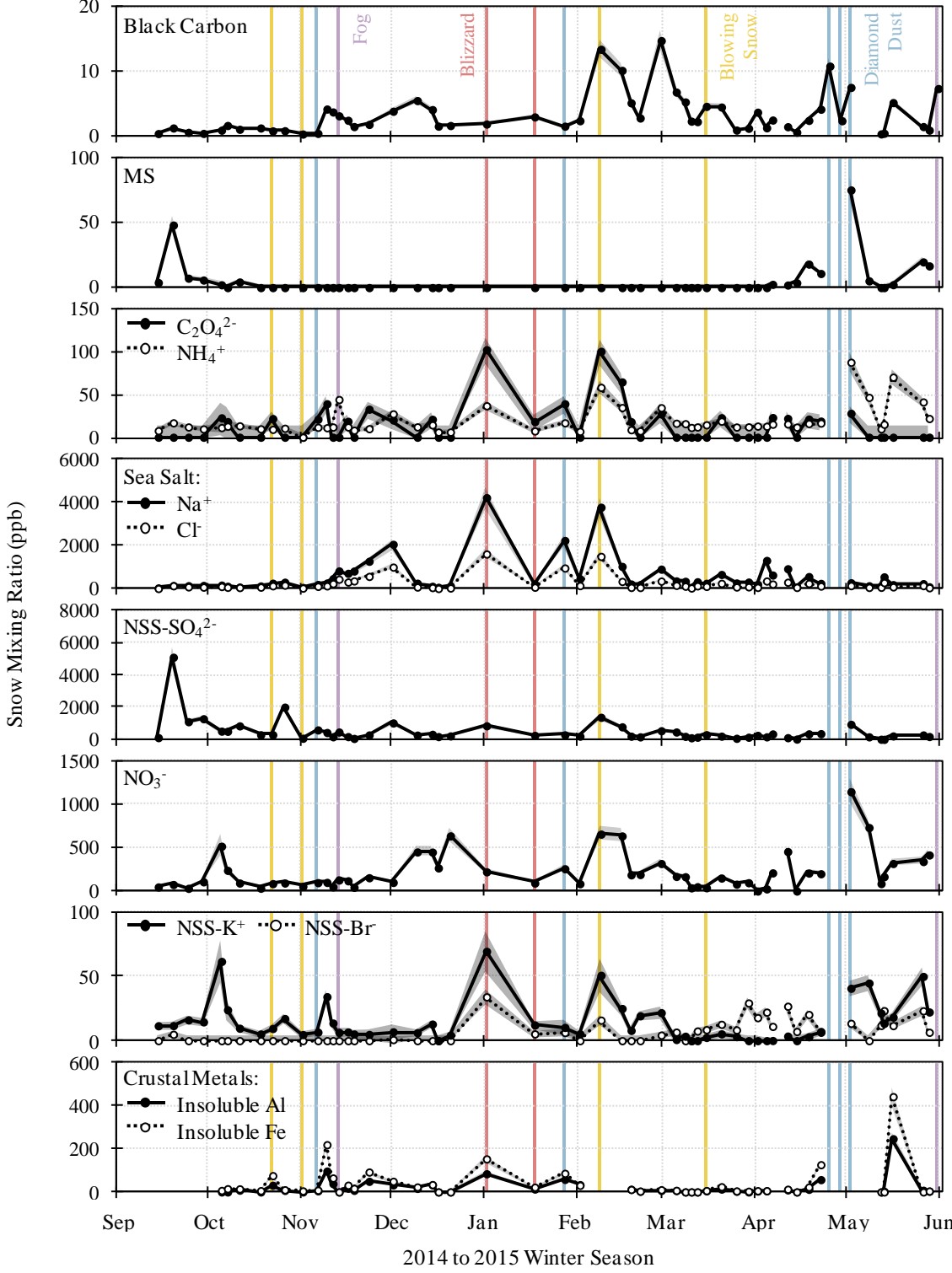

**Figure 1: Measured snow mixing ratio (line) and uncertainty (shaded area) of key analytes over 2014 to 2015 campaign.**

A review of existing Arctic snow measurements found the measured median mixing ratios to fall within expected ranges (see supplemental Table S5 for details); however, it should be noted that the referred data represent a variety of collection and analysis techniques. In general, measurements of this campaign showed salt species and non-crustal metals to be at the lower end of the typical range while $SO_4^{2-}$ and $NO_3^-$ were at the higher end. A limited number of seasonal snow collection

campaigns were available for comparison of the observed seasonal trend in analytes. The winter peak in BC was similar to that observed by Davidson et al. (1993); however, spring values observed in this campaign were higher than previously seen. The observed seasonal trend in major ions was generally consistent with existing literature (Davidson et al. 1993; Toom-Sauntry and Barrie, 2002; Dibb et al., 2007). Specifically, a winter peak in sea salt, fall/spring peaks in MS, and a winter peak in $NO_3^-$ are all typical. However, the fall peak observed in the NSS-$SO_4^{2-}$ mixing ratio and spring peak in $NO_3^-$ were

unlike seasonal trends observed previously (Davidson et al. 1993; Toom-Sauntry and Barrie, 2002). Unusual weather events, as noted by the operators, are highlighted in Figure 1; however, no obvious relationship was observed with snow measurements, with the exception of blizzard and high wind conditions in January and February which were associated with elevated mixing ratios for several chemical species. Atmospheric measurements are provided in the supplemental Table S7. As observed for snow, most atmospheric analytes experienced a winter high. The fall/spring peaks in MS and notable fall

peak in $SO_4^{2-}$ persisted in both snow and atmospheric measurements.

### 3.2 Factors Influencing Snow Scavenging and Deposition

### 3.2.1 Effective Deposition Velocity

As discussed above, the collected snow samples provide information on the total deposition of material to the surface over a given time and area. In order to elucidate the mechanisms controlling this bulk deposition, a simplistic model for flux, Eq.

(1), was adopted to describe the measured deposition:

$$F_{S,total}{}^{ij} = C_A{}^{ij} \, v_{d,eff}{}^{ij} \, , \tag{1}$$

where $F_{S,total}{}^{ij}$ is the flux deposited to snow of the $j^{th}$ analyte over the $i^{th}$ period; $C_A{}^{ij}$ is the arithmetic average atmospheric concentration; and $v_{d,eff}{}^{ij}$ is the effective deposition velocity.

The measured snow flux ( $F_{S,total}$ ) represents the total deposition by wet and dry mechanisms. Thus, the effective deposition

velocity ($v_{d,eff}$) encompasses the variety of aerosol and meteorological properties controlling deposition efficiency and the relative importance of each deposition pathway: dry, wet in-cloud, and wet below-cloud scavenging. This effective velocity is akin to a typical deposition velocity; however, it encapsulates the bulk movement of material by combining the dry deposition velocity and the wet deposition efficiency as an equivalent velocity. Thus, this parameter can be used to provide a holistic view of Arctic deposition. Use of the effective deposition velocity also avoids the uncertainties of estimating the split

between dry and wet deposited mass. A caveat to this analysis is that the three deposition mechanisms relate to different atmospheric concentrations, a gradient which is not necessarily captured when the ground-level atmospheric concentration ( $C_A$ ) is used to calculate the effective deposition velocity: dry deposition affects the lower atmosphere, in-cloud scavenging

the cloud layer, and below-cloud scavenging the full below-cloud atmospheric column. Previous observations of vertical profiles in the Arctic have shown notable variability with altitude (Hansen and Rosen, 1984; Leaitch et al., 1989; Spackman et al., 2010; Brock et al., 2011; Sharma et al., 2013). So, the calculated effective velocity includes an intrinsic variability dependent on the vertical atmospheric profile of each analyte.

Effective deposition velocities were calculated for chemical species measured in both snow (SP2 and IC) and atmospheric (SP2 and Hi-Vol) samples. Figure 2 shows effective deposition velocities calculated as the ratio of total summed snow flux and average atmospheric concentration measured over the same period. Both a six-day resolution, as per the Hi-Vol sampling frequency, and monthly resolution are provided. The calculated effective deposition velocities ranged from 0.001 to 10 cm/s (0 to 16 cm/s with uncertainty). Episodic and monthly peaks are observed in Figure 2 for each analyte. The

variance in deposition observed by composition and temporally is discussed below.

### 3.2.2 Variation in Deposition by Composition

Monthly effective deposition velocities were used to contrast deposition mechanisms by aerosol composition. A monthly resolution provides insight into the general deposition regime of each analyte, highlighting the impact of bulk deposition characteristics rather than event-specific variability. The variability between aerosol of different composition and the

influences of seasonal changes within the Arctic system are simpler to identify without the interference of variability between event-specific conditions. A monthly analysis also facilitates future comparison with modelled results which may not replicate individual events. January and February, 2015, were excluded from the monthly analysis because blizzard and high wind conditions were believed to have caused significant losses of snow from the snow tables during these months (based on operator reports), which would lead to underestimation of these snow flux values. The effective deposition

velocity is best suited to analysis across periods of equal length and precipitation volume, since both of these parameters are inherently included when the wet deposition efficiency is converted to an equivalent deposition velocity. With the exception of January and February, the total monthly snow precipitation over the campaign was relatively constant, with a relative standard deviation of 20%.

Figure 3 shows that the typical deposition characteristics varied by analyte, with median effective deposition velocities

ranging from 0.03 to 1.1 cm/s; however, error bars describing the combined uncertainty of the snow and atmospheric measurements show that the range of calculated effective deposition velocities of each analyte have considerable overlap. The calculated monthly effective deposition velocities showed a relative standard deviation of 60 to 150% across the measured analytes, while measurement-based uncertainty was estimated as only ±35%, indicating a significant impact of aerosol state or properties on deposition.


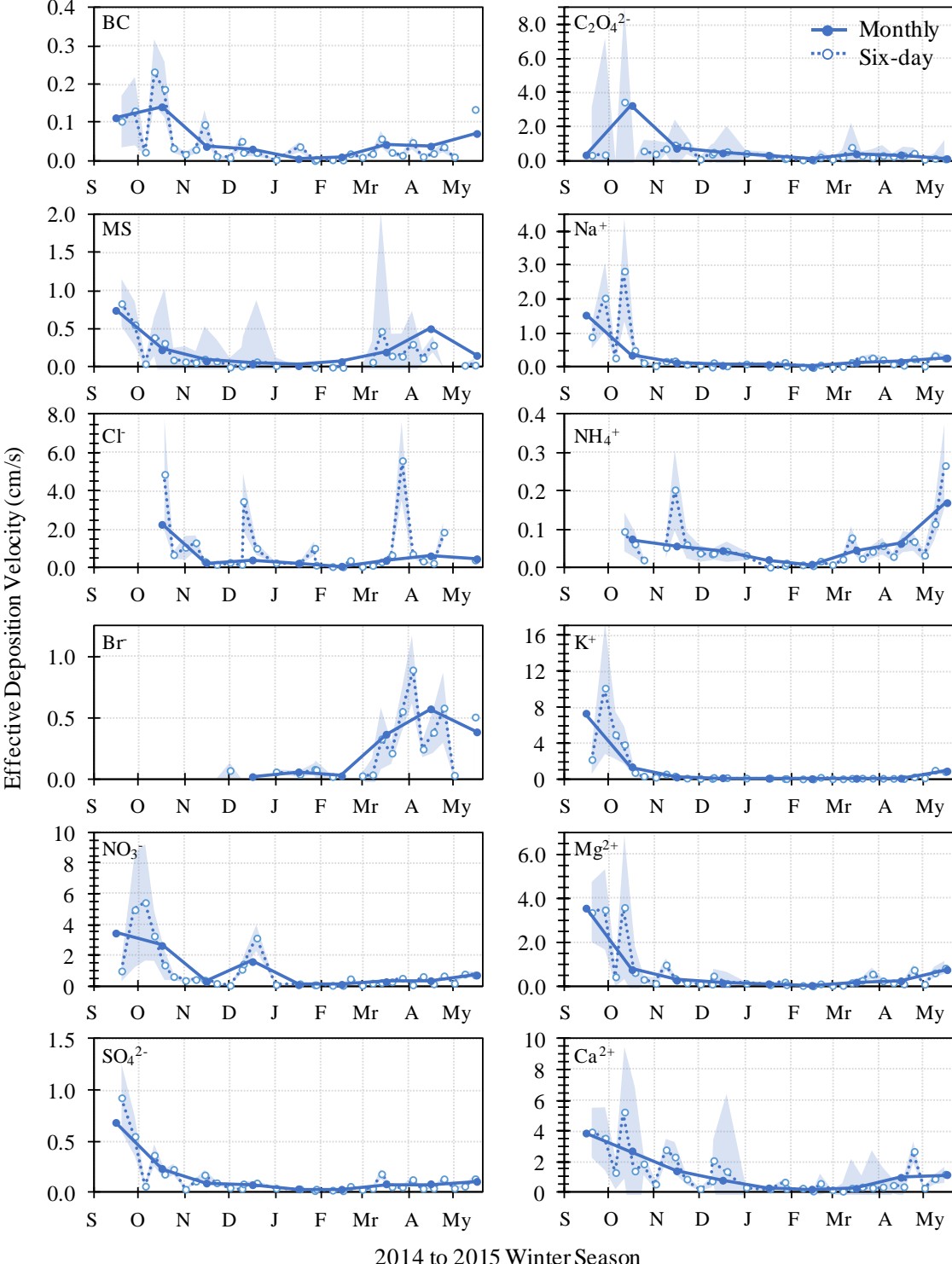

**Figure 2: Effective deposition velocity at monthly (solid) and approximately 6-day (dashed) frequencies with 6-day uncertainties (shaded area). Missing values indicate periods with snow and/or atmospheric measurements below detection limits.**

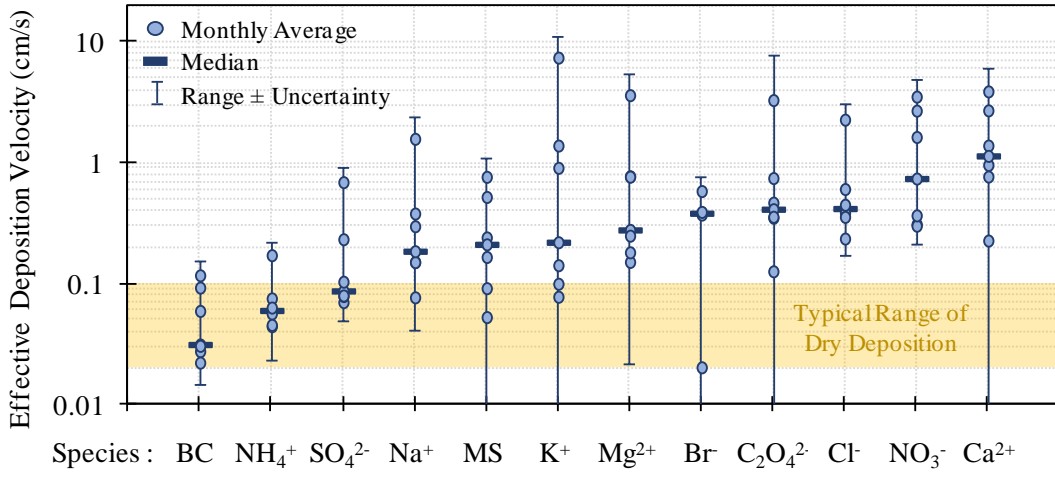

**Figure 3: Monthly effective deposition velocities by composition (points, excluding January and February). The median of each analyte (bar) and full range with uncertainty (error bar) are also shown. Also shown is the typical range of dry deposition velocity for accumulation mode particles to snow by others (Davidson et al., 1987; Petroff and Zhang, 2010).**

The variability observed between analytes may be the result of variations in aerosol properties. First, the measured chemical species differ in terms of dominant phase: BC, $NH_4^+$ (ammonium), $SO_4^{2-}$, $Na^+$, $K^+$, $Mg^{2+}$, $C_2O_4^{2-}$ (oxalate), and $Ca^{2+}$ are typically observed predominantly in the particle phase within the Arctic, while MS, $Br^-$, $Cl^-$, and $NO_3^-$ and their associated precursors can have appreciable gas-phase portions (Barrie and Hoff, 1985). In general, analytes considered to be dominantly particle-based were seen to exhibit lower deposition velocities than those which may have existed as gases. This

observation is supported by a detailed study of sea salt species. The Cl in salt particles has been previously observed to partition to the gas phase over winter months (e.g., Barrie and Hoff, 1985; Toom-Sauntry and Barrie, 2002; Quinn et al., 2009); thus, Arctic salt particles often show a Cl deficit while gaseous Cl shows enhancement. Such a process was corroborated by comparison of the atmospheric Hi-Vol and snow IC measurements for $Na^+$ and $Cl^-$. Both atmospheric and snow measurements showed high correlation of $Na^+$ and $Cl^-$ analytes, 0.92 and 0.99, respectively. Yet, the ratio of $Cl^-/Na^+$

differed between measurement mediums with a higher proportion of $Cl^-$ observed in snow ($Cl^-/Na^+$=2.49 mass/mass) than in atmosphere ($Cl^-/Na^+$=1.36). Compared to the typical marine ratio ($Cl^-/Na^+$=1.795; Pytkowicz and Kester, 1971), the atmosphere showed a deficit in $Cl^-$ while snow showed enhanced $Cl^-$ (see supplemental Figure S4). It is expected that the Hi-Vol measurement technique would collect predominantly atmospheric particles, while snow would scavenge both gaseous and particulate aerosol. Thus, the enhanced $Cl^-$ in snow above that of sea salt indicates that Cl partitioned to the gas phase,

that both phases deposited to snow, and that gas-phase Cl was scavenged to snow preferentially. Assuming all $Na^+$ and $Cl^-$ originated as sea salt particles, the $Cl^-$ mass from each phase can be estimated in the atmosphere and snow. Based on these estimated proportions in the snow and atmosphere, median effective deposition velocities were calculated as 0.16 and 0.40 cm/s for the particle and gas phases, respectively. Comparison of these velocities indicates an 86% enhancement in gas-

phase deposition relative to that of the particle phase. Thus, gaseous scavenging may contribute to the enhanced bulk deposition observed in MS, Br$^-$, Cl$^-$, and NO$_3^-$.

Particle nucleation affinity may also be a significant contributor to the observed differences in bulk deposition. The lowest velocities were observed for BC, NH$_4^+$, and SO$_4^{2-}$, which largely fell within the typical range of deposition velocities seen for dry deposition: 0.02 to 0.10 cm/s for typical accumulation mode Arctic particles (Davidson et al., 1987; Petroff and Zhang, 2010), as highlighted in Figure 3. In contrast, the monthly velocities of other particle-dominated chemical species, Na$^+$, K$^+$, Mg$^{2+}$, C$_2$O$_4^{2-}$, and Ca$^{2+}$, all fell above the typical dry deposition range. Particles containing these analytes, as well as NH$_4^+$ and SO$_4^{2-}$, have been previously suggested to act as better precipitation nuclei than BC (e.g., Zobrist et al., 2006; Hoose and Möhler, 2012; Farmer, Cappa, and Kreidenweis, 2015); thus, their enhanced velocities might be attributed to in-cloud scavenging. Thus, dry deposition may be the dominant deposition mechanism in some measurements while others show enhanced deposition which likely reflects an increased contribution of wet deposition processes. The observed difference in deposition velocities also implies that BC/SO$_4^{2-}$/NH$_4^+$-containing particles are to a significant extent externally mixed from these other constituents.

Furthermore, salt and crustal particles may experience enhanced deposition since they typically consist of coarser particles than BC, SO$_4^{2-}$, and NH$_4^+$. Specifically, coarse mode particles can exhibit dry deposition velocities to snow up to 0.6 cm/s and below-cloud scavenging efficiencies enhanced over accumulation mode particles by a factor of 10 (Zhang and Vet, 2006; Petroff and Zhang, 2010). The enhancement of Ca$^{2+}$ above other crustal-related analytes is unexpected. This cannot be satisfactorily explained without further study; however, the phenomenon may be connected to the dissimilar sources of Ca-rich mineral dust to that of other crustal species suggested by other studies (Banta et al., 2008). In addition to aerosol phase, nucleation affinity and size as discussed above, particle coating could also impact the scavenging and deposition process. However, the potential influence of coatings on the observed velocities cannot be addressed with the available information.

### 3.2.3 Temporal Variability of Deposition

Shared temporal trends in effective deposition velocity can be observed in Figure 2. A general trend of heightened deposition in the fall and spring can be observed across all analytes. In particular, BC, Na$^+$, Mg$^{2+}$, and Ca$^{2+}$ can be seen to share a similar seasonal and episodic trend, with Pearson's correlation coefficients of 0.7 to 0.9 (comparing six-day resolution, excluding January and February, 2015, p-values <0.001). Episodic peaks in Mg$^{2+}$ and Ca$^{2+}$ also show some similarity to SO$_4^{2-}$ (correlation of 0.70-0.85, p-value <0.001). The trend of NH$_4^+$ is more difficult to distinguish as September measurements were below detection limit; however fall and spring peaks are suggested. A more pronounced fall peak was observed in SO$_4^{2-}$, Na$^+$, K$^+$, and Cl$^-$, though their episodic peaks differ. Although MS is distinguished by a spring peak, MS and SO$_4^{2-}$ share similar episodic peaks with a six-day correlation of 0.86 (p-value <0.001). The seasonal trends of Br$^-$, C$_2$O$_4^{2-}$, and NO$_3^-$ are more distinct: Br$^-$ exhibits a broad spring peak, C$_2$O$_4^{2-}$ a short October peak, and NO$_3^-$ a broad fall/winter peak. Episodic C$_2$O$_4^{2-}$ showed moderate correlation with BC, Na$^+$, Mg$^{2+}$, and Ca$^{2+}$ (coefficient=0.5-0.7, p-value<0.02) and NO$_3^-$ episodes showed correlation with K$^+$ (coefficient=0.83, p-value<0.001). There are some peak events shared across several

analytes, for example, an early October peak is observed in most chemical species. While the differing magnitudes of effective deposition velocities observed across analytes implies separate scavenging and deposition, these shared temporal trends indicate that the externally mixed particles and gaseous chemical species are subject to similar temporal influences controlling deposition.

Several factors controlling deposition experience seasonal variations, which may have contributed to the observed inter-monthly variability. Six properties of the Arctic system with seasonal trends were considered as possible influences on the observed velocity trend: precipitation, temperature, mixing height, cloud height, dominant aerosol source region, and sunlight availability, as shown in Figure 4. The precipitated snow-water equivalent depth was calculated from the snow mass and table area of each sample. Temperature was monitored at local ground-level meteorological stations over the campaign

(supplemental Table S9) and sunlight estimated from location and time of year. The dominant aerosol source of each month was described using the southern limit to transport and mixing/cloud heights were estimated from radiosonde data, as described above.

Temperature, transport, and sunlight can be seen to follow similar seasonal trends with fall/spring peaks. Precipitation, mixing height, and cloud height exhibit episodic peaks and a less significant seasonal trend (relative standard deviation intra-

monthly was a factor of 1.5 to 2 times higher than inter-monthly, whereas these values were approximately equal for temperature, transport, and sunlight). When compared to effective deposition velocities, BC, $SO_4^{2-}$, $Na^+$, $Mg^{2+}$, $NO_3^-$, and $Ca^{2+}$ show better correlation with temperature than the other meteorological conditions shown in Figure 4, with Pearson's correlation coefficients above 0.5 (excluding January and February, 2015, p-value <0.02). Specifically, $SO_4^{2-}$, $K^+$, $Mg^{2+}$, and $Ca^{2+}$ had high correlation coefficients with temperature of 0.6-0.7 (p-values <0.002). Medium to high correlations were

observed between mixing height and $SO_4^{2-}$, $Na^+$, $K^+$, $Mg^{2+}$, and $NO_3^-$, with coefficients of 0.4-0.7 (highest correlation for $K^+$ at 0.76) and p-values below 0.05. Bromide deposition showed high correlation with sunlight (coefficient: 0.6, p-value: 0.001). In contrast, $NH_4^+$ and MS only showed weak correlation with the described meteorological parameters (maximum coefficients of about 0.4 with transport and mixing height, respectively). In contrast, $C_2O_4^{2-}$ and $Cl^-$ did not show strong correlations with any of the described meteorological parameters (coefficients below 0.2). Overall, temperature showed the

best correlation with deposition velocities for most analytes, followed by mixing height. Thus, changes in temperature may be linked to seasonal changes in the dominant scavenging mechanism of several analyzed species. Specifically, it is hypothesized that the increased presence of mixed-phase clouds with warmer temperatures may have enhanced wet deposition via in-cloud scavenging due to CCN activity for those analytes expected to exist predominantly in the particle phase: BC, $NH_4^+$, $SO_4^{2-}$, $Na^+$, $K^+$, $Mg^{2+}$, $NO_3^-$, and $Ca^{2+}$. This aligns with seasonal trend in $SO_4^{2-}$ deposition observed by

Davidson et al. (1985b and 1987) and the suggested link to deposition mechanism.

The effective deposition velocities for warmer and colder months were separated using a -20 °C cut-off between ice-cloud dominated periods, November to April (N/D/Mr/A), and months with a significant potential for mixed-phase clouds, September, October, and May (S/O/My). Radiosonde observations show temperatures at typical cloud heights usually above

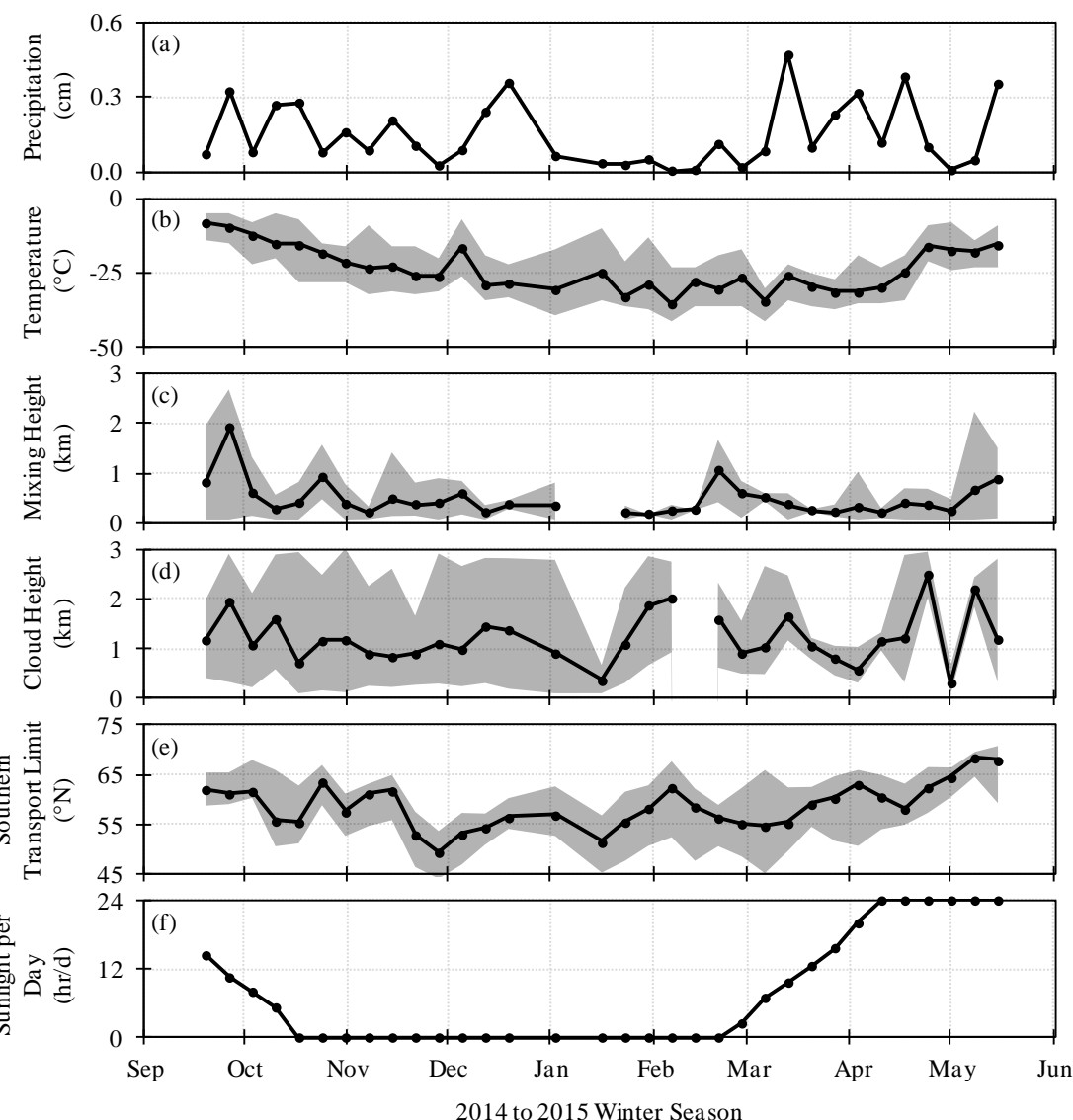

**Figure 4: Seasonal variation in precipitation (a), temperature (b), mixing height (c), cloud height (d), transport (e), and sunlight (f). Precipitation and sunlight are presented as the average per snow sample collection period. Temperature, mixing/cloud height, and southern transport limit are presented as the average for each collection period (line) along with the full range (shaded area).**

5   -20 °C in S/O/My and below -20 °C in N/D/Mr/A, supporting the mixed-phase cloud hypothesis. Comparison of the effective deposition velocities delineated by these periods minimized the influence of precipitation volume as it was relatively constant: an average of 12 mm/month in S/O/My and 18 mm/month in N/D/Mr/A. However, the influence of aerosol source may differ by period given their distinct source profiles: long-range transport dominated in N/D/Mr/A, and local transport dominated in S/O/My. Figure 5 depicts the range of effective deposition velocities calculated for each analyte

10   over these two periods (again excluding January and February). Bromide was excluded from this analysis since it was below detection limit in the snow and/or atmospheric measurements from September to November.

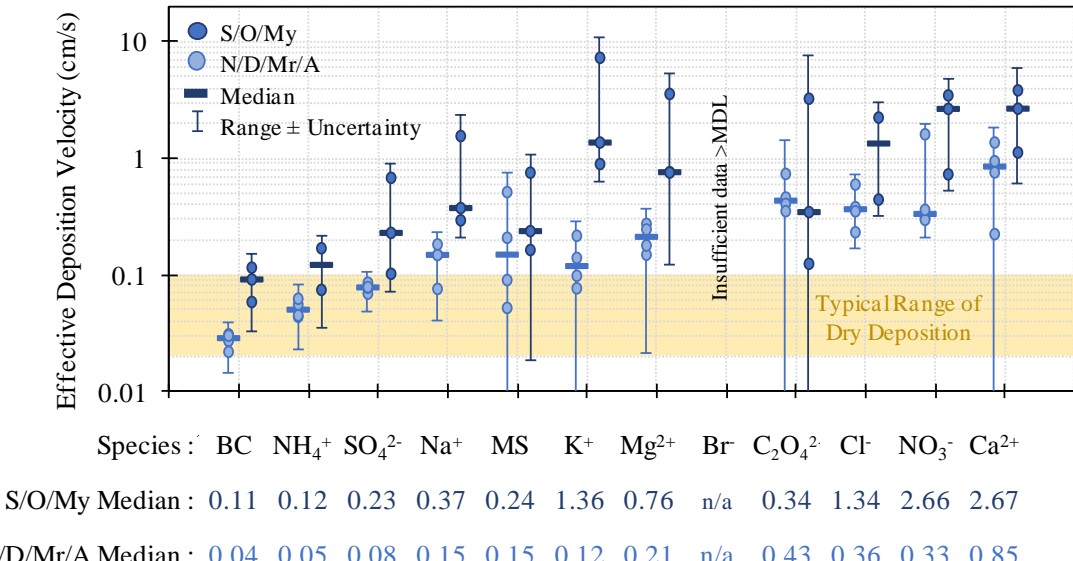

| Species : | BC | NH$_4^+$ | SO$_4^{2-}$ | Na$^+$ | MS | K$^+$ | Mg$^{2+}$ | Br$^-$ | C$_2$O$_4^{2-}$ | Cl$^-$ | NO$_3^-$ | Ca$^{2+}$ |
|---|---|---|---|---|---|---|---|---|---|---|---|---|
| S/O/My Median : | 0.11 | 0.12 | 0.23 | 0.37 | 0.24 | 1.36 | 0.76 | n/a | 0.34 | 1.34 | 2.66 | 2.67 |
| N/D/Mr/A Median : | 0.04 | 0.05 | 0.08 | 0.15 | 0.15 | 0.12 | 0.21 | n/a | 0.43 | 0.36 | 0.33 | 0.85 |

**Figure 5: Effective deposition velocities split by season. The effective deposition velocities are separated into two time periods: warmer months S/O/My = September/October/May, and colder months N/D/Mr/A = November/December/March/April.**

With the exception of C$_2$O$_4^{2-}$, all analytes showed a larger median effective deposition velocity for the warmer S/O/My months than the colder N/D/Mr/A months. Although insufficient data are available to confirm the statistical significance for each individual analyte, the combined normalized dataset of velocities showed significant enhancement during S/O/My using the ANOVA test (p-value 4.9×10$^{-8}$). Specifically, marked enhancement was seen for BC, NH$_4^+$, SO$_4^{2-}$, Na$^+$, K$^+$, and Mg$^{2+}$. The cold-month deposition of these analytes can largely be described by dry deposition alone with velocities below 0.1 cm/s for analytes expected to be dominated by accumulation mode particles and 0.6 cm/s for those expected to include significant coarse-mode mass; however, their warm-month effective deposition velocities were a factor of 2 to 12 higher.

The unexpected discrepancy in the deposition of Ca$^{2+}$ and Mg$^{2+}$ given their typically similar origins, as discussed above, was also observed when comparing median warm and cold month periods. Both Mg$^{2+}$ and Ca$^{2+}$ are common to sea salt and crustal origins but their non-sea salt (NSS) portions can be estimated based on typical sea salt ratios with Na$^+$ (Pytkowicz and Kester, 1971). The NSS-Mg$^{2+}$ and NSS-Ca$^{2+}$ showed similar behaviour over the warm months, with average velocities of 2.4 and 2.6 cm/s, respectively, but a larger difference in the cold months, with average velocities of 0.4 and 1.0 cm/s, respectively. No explanation of this discrepancy can be supported without further study; however, it appears that Ca$^{2+}$ and Mg$^{2+}$ exist as externally mixed Ca-rich and Mg-rich crustal particles which are subject to differing deposition processes, particularly in dry deposition-dominated colder months.

Thus, the observed effective deposition velocities suggest that most analytes and particularly those expected to exist primarily as particle phase were preferentially scavenged during the warmer S/O/My months, possibly due to the presence of mixed-phase clouds and the associated CCN activation of these chemical species or enhanced below-cloud deposition of those compounds typically associated with larger particles. However, the change in source profile typically experienced

during these months along with other seasonal changes in aerosol processing and altitudinal distribution might have also contributed to the observed S/O/My enhancement. In particular, records of volcanic activity show that the Icelandic volcano Bárðarbunga was active August, 2014 through February, 2015 (Global Volcanism Program, retrieved March 2016 from http://volcano.si.edu/), which likely contributed to a shift in the dominant source and scavenging-related properties of $SO_4^{2-}$
over the campaign that would not be representative of a typical year. Thus, the $SO_4^{2-}$ observations of this campaign, especially the peak snow mixing ratio seen in the fall, may not reflect a seasonal trend for typical Arctic Haze. The overlapping warm/cold-month ranges observed for $Cl^-$, MS, $C_2O_4^{2-}$, and $NO_3^-$ suggest that the deposition of these chemical species was more strongly driven by factors other than nucleation, such as gas-phase partitioning or other aerosol aging processes. For example, enhanced deposition of MS and $Br^-$ was observed as early as March and April, which could imply
that their deposition was impacted by changes in the atmospheric processing of these chemical species during polar sunrise.

**4 Conclusions**

To help characterize the chemical state of the rapidly changing high Arctic, an intensive campaign of fresh snow sampling at Alert, Nunavut, was completed and snow quantified for a broad suite of analytes. Comparison of these snow measurements with coincident atmospheric measurements allowed estimation of monthly effective deposition velocities describing the total
dry and wet deposition in the range of about 0.02 to 8 cm/s. The calculated effective deposition velocities for several measured chemical species resemble those expected for dry deposition alone, suggesting that dry deposition may be the dominant removal mechanism, especially for winter scavenging of BC, $NH_4^+$, $SO_4^{2-}$, $Na^+$, and $K^+$. Enhanced deposition during September, October, and May suggests that wet deposition may increase in importance during these warmer months, possibly due to the presence of mixed-phase clouds and the associated scavenging of crustal, salt, and $SO_4^{2-}$ species as CCN;
however, other factors such as changes in the dominant aerosol source profile may also contribute to the observed trend. Comparison of salt species measurements in the Arctic snow and atmosphere suggested that Cl experiences significant gas-phase partitioning and that this gas phase may be preferentially scavenged. Such gas-phase scavenging may contribute to the enhanced deposition of MS, $Br^-$, $Cl^-$, and $NO_3^-$ observed relative to BC, in conjunction with other aerosol processing differences. The low deposition velocity of BC-containing particles is consistent with those particles being externally mixed
from more soluble species and having a low cloud nucleation efficiency. Given the rarity of temporally-refined and broadly speciated Arctic snow sampling campaigns, measured deposition magnitudes and insights on deposition mechanisms such as these are valuable for future model validation.

**Author Contribution**

Organization of the snow collection campaign was led by S. Sharma with the assistance of A. Platt and sample collection by M. Elsasser. Snow SP2 analysis was completed by J. McConnell and N. Chellman, snow IC analysis lead by D. Toom with

the assistance of A. Chivulescu, and snow ICP-MS by K. Macdonald with the assistance of Y. Lei. Analysis of radiosonde data was completed by D. Tarasick. Ambient atmospheric monitoring of inorganic aerosols was completed by D. Toom and monitoring of BC by S. Hanna with the assistance of A. Bertram. FLEXPART simulations were completed by H. Bozem and D. Kunkel with data analysis assisted by K. Macdonald. Data interpretation was led by K. Macdonald with input and

comments by all authors. G. Evans and J. Abbatt provided oversight for the project, including input on the manuscript.

## Competing interests

The authors declare that they have no conflict of interest.

## Acknowledgements

Funding of this study was provided as part of the Network on Climate and Aerosols Research (NETCARE), Natural Science

and Engineering Research Council of Canada (NSERC), the government of Ontario through the Ontario Graduate Scholarship (OGS), and Environment and Climate Change Canada. This project would not have been possible without the collaboration of many skilled individuals: Richard Leaitch at Environment Canada; and Catherine Philips-Smith and Cheol-Heon Jeong at the University of Toronto.

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
