# Peer review of "Observations of Atmospheric Chemical Deposition to High Arctic Snow"

_Atmospheric Chemistry and Physics, 2016_

## Referee Comment (RC1) · Anonymous Referee #1 · 6 Dec 2016

This manuscript presents results from a 9-month long campaign that collected samples of freshly fallen snow as soon as possible after each deposition event that occurred at Alert from mid-September 2014 through mid-May 2015. These samples were obtained using a technique that the authors assert allowed estimation of combined wet and dry deposition over the interval between each collection, and were analyzed for BC, major ions, and a large number of trace elements. The authors also use measurements of the concentration of aerosol-associated BC, ions and a smaller number of trace elements made as part of the baseline suite by station staff to explore the transfer of impurities from the atmosphere to the snow.

The data set is quite rich and interesting, however I feel that the interpretation and synthesis does not exploit the data set well. A major problem is the decision to define and rely on the "effective deposition velocity" (EDV)for nearly all of the discussion. The

authors acknowledge on page 2 that wet deposition is dependent on a wide range of factors, many of which change from day to day, such that previous theories have never found it useful to consider a wet deposition velocity. Then on page 9 they point out additional reasons that such a construct is bound to obscure critical details relating to the processes removing aerosol associated (and gas phase) impurities from the atmosphere to the snow surface, without presenting a strong argument why ignoring these details is reasonable.

To my mind this bad decision is compounded by focusing on monthly values of the estimated EDV as the highest temporal resolution (but only shown in Figure S3). In the actual manuscript one of the key figures (Fig. 2) shows estimated EDV for BC and ions, but only as box and whisker plots summarizing the full campaign (minus Jan and Feb which were discarded due to concerns about data quality, more on that later). Another key figure (Fig. 4) shows monthly EDV, but lumps all of the different impurities together, obscuring significant differences that are evident in Fig. S3 with no explanation. Likewise, Fig. 5 does include separate EDVs for BC and the ions, but presents statistical summaries for a 3 month warm season compared to 4 month cold season (again, omitting Jan and Feb).

The authors should establish a stronger case for the utility of the EVD, and it should be calculated and presented at the 6-8 day resolution of the aerosol sampling. A revised version of Fig S3 (with approximately 4 times as many data points) should be in the main text, informing most or all of the discussion. In the detailed comments below I will work through the manuscript page by page with a mixture of editorial and content comments. The latter will be based on statements in this draft in relation to the figures that are now included, hence may be less relevant if a revised manuscript focuses on the higher resolution that the data set could provide.

Before turning to detailed comments I need to raise one other general objection to this draft. The authors claim repeatedly that this data set is unique by virtue of the large number of impurities quantified, that their sampling allowed fluxes to be determined,

and that sampling captured all (and only) fresh snow events. I would like to point out that the DGASP campaign at Dye 3 in south Greenland took a very similar approach back in 1988-89, though many of the snow samples were not collected in a way that allowed net accumulation (hence flux) to be quantified. This shortcoming of DGASP sampling was addressed in a long series of summer time campaigns at Summit, Greenland (1989 to at least 2003) that included daily sampling to quantify both concentration and the mass of impurity/unit area in the surface layer of snow (see series of papers by Bergin et al. from the mid 90's with "Flux" in their titles as examples), and was maintained for year round campaigns at Summit in 1997-1998 and 2000-2002 (Dibb et al., 2007). With better hindsight, the authors may want to rephrase the statements in this draft made on page 1, lines 20-21; page 2, lines 1-2; page 3, lines 2-3; page 3, lines 20-22; page 6, line 12; page 7, lines 3-6 (particularly this one!); page 16, lines 3-5.

Detailed comments referenced to page/line #(s)

1/16 as noted above "effective deposition velocities" not established, and maybe not useful, a strong case needs to be made later that the concept does have utility, or these calculated values should not be highlighted in the abstract

2/25 dependent on snow type

2/27-29 not sure what the message of sentence starting "The efficiencies..." is supposed to be

2/30 Last sentence is quite true, and for pretty good reasons. Why are you bucking the trend?

3/11 Based on the picture in Fig S2 I strongly suspect that the snow table had a major problem with undercatch if there was any wind at all (not just high winds causing blowing snow). It would have been prudent to check this fairly frequently through the campaign by determining the cumulative accumulation of snow on the ground near the table. At a minimum, determination of SWE from either a small pit or a simple core

could have been compared to a summation of the SWE for all of the layers collected off the table. Better would have been to check the inventory (mass/unit area) from snow surface to the ground of selected impurities at some interval (at least monthly), by collecting and analyzing the full column of accumulated snow. In the absence of this kind of validation, you probably need to increase the uncertainty estimate for your impurity fluxes.

4/16 What were the QC runs?

4/31 How, exactly, was "mass recovery precision" determined? Sounds like some kind of yield tracer was added, but no details given.

5/4-6 When was this "internal standard" added? Before or after digestion? Did it also serve as the yield tracer to track recovery?

5/11 in a class 100

5/25-26 Are U. Toronto MA thesis publicly available? If not, are they citable in ACP?

6/14-24 There is a sizeable literature about the problems with surrogate surfaces for the collection of both dry and wet deposition. Some of that should be cited here.

6/24-25 May want to include comparison between cumulative depth, or cumulative SWE probably better, on the table and at the met station. This would allow the reader some insight into how bad table undercatch may have been (or persuade you to make some correction).

6/27 Flux per day over 2-18 days makes no sense for a wet deposition event likely lasting just hours. Understood that you can not separate wet and dry in this data set, but this contributes to discomfort with the calculated EVDs.

8 (Fig 1) Why are there gaps of different lengths and at different times in the time series of BC, ions, and crustal metals? According to section 2.1, anytime one bottle was filled there should have been 3 others (presumably for the different analyses). Problems

causing gaps need to be noted and enumerated (probably better in 2.1 than here).

9/13-16 Why are the much more numerous flights from the 4 NOAA AGASP campaigns or the 5-6 NSF TOPSE flights not included here? My impression is that some of these showed significant gradients in the lowest 3 km. (Note that the special issue of Atmos Environ on DGASP also includes AGASP3 papers and provides access to the earlier ones.)

9/16-18 Somewhat bad form to use unpublished data to support your position, but beyond that, flights in April are not really informative about stability and stratification in the winter. Recall, you are asserting these data are unique because they include the cold season (see later comments related to your Figs 4 and 5).

9/21-27 As noted, assuming you can make a better case for utility of EVDs, I do not like the use of monthly averages here and do not think the final sentence is valid justification. There are a lot of papers focused on the POLARCAT airborne campaigns where global and regional CTMs attempted to simulate day by day, plume by plume in some cases, and more should be expected as part of recent and future campaigns. Modelers are striving to improve their simulations, so if experimentalists provide high resolution records they welcome the challenge of simulating them, however poorly the first round of comparisons turn out.

9/28 As noted earlier, summarizing 7 months in box and whisker plot makes little sense to me.

10/10-13 To me, the most puzzling aspect of Fig 2 is why Ca is so much higher than Na, Cl and Mg. This is not even mentioned until page 15 (with no plausible explanation). Similarly, suggesting here that BC, NH4 and SO4 seem dominated by dry dep causes some tension with later suggestion that they are serving as CCN (or at least being efficiently scavenged) in S/O/My (Fig 5)

11/5-6 Not so sure this is true. The Hi-Vol sampling at Alert used to use cellulose filters

(Whatman 41s) which do collect acids with decent efficiency. And for a filter exposed for 6-8 days, seasalt and dust can build up to the point where acids may be collected quantitatively.

10/7 to 11/8 Chloride deficit in MBL aerosol and excess in Arctic snow (especially in summer) has been well documented by many others in addition to Toom-Sauntry and Barrie, 2002. This section seems too long for significance of the point being made, but also lacking references to prior work.

12/14 enhanced deposition

12/18-21 As pointed out in first sentence of this paragraph, everything shown in Fig 2 has higher EVD than BC, so what is so special about these 4? They are not as enhanced as Ca in the median, and max EVD for K, Mg, C2O4 and Ca are all higher than the max for MSA, Br and Cl. I agree that the 4 you highlight can have appreciable gas phase, but that is not something discovered or even really supported by Fig 2.

Pages 12-14 (Figs 4 and 5) As noted at top, I like that 4 at least shows monthly resolution, but don't like lumping all impurities, while 5 is nice in showing BC and all ions, but is averaging over too much time.

More importantly, I think both are missing key point (which calls the value of EVD into question).

Perhaps the most important feature of the Arctic is the huge seasonal contrast in daylight, which leads to big contrasts in energy balance and atmospheric stability. In the winter the atmosphere is extremely stable with weak vertical mixing. Atmospheric impurities that get into the BL tend to be trapped there, and concentrations can be high. As things warm up the BL deepens and concentrations tend to decrease close to the surface. This is clearly reflected in Table S8 which shows very low concentrations of BC, Cl, C2O4, Na, K, Mg in SOMy compared to the colder months. Pattern is not so strong for NO3, NH4 and SO4, but the averages for the 3 warm months are quite low

compared to the 6 (or 4) cold ones.

Part of this may be transport related, and local biogenic emissions may account for the weaker trend in SO4 and opposite trend in MSA, but dilution into a much deeper BL is probably an important factor.

As a thought experiment, given constant monthly accumulation of snow, and constant burden of impurities in the BL, and assuming complete cleansing of the BL over each month, but a 5-10 fold deeper and well mixed BL in warm months compared to cold ones, what would the EVDs look like.

Constant monthly flux to snow divided by 5 (or 10) fold lower aerosol concentration measured at surface would yield 5 (or 10) times larger EVD in the warm months. Clearly, the story is not so simple. But, is it really plausible that scavenging in mixed phase clouds is the major factor accounting for increase in BC and all the ions (which should all behave differently in cloud)?

See again the suggestion to include a higher resolution version Fig S3 as the focus of discussion, and exploit the differences to gain insight.

---

## Referee Comment (RC2) · Anonymous Referee #2 · 6 Jan 2017

This manuscript describes measurements of the chemical composition of freshly precipitated snow from Alert, Canada and relates those snow measurements to aerosol measurements to derive deposition velocities for various species. The manuscript reads well, and describes a unique contribution to the literature and thus is appropriate, after revision for eventual publication in ACP. The work also uses a comprehensive set of multi-species analyses (BC, IC, ICP-MS) that are a valuable expansion on prior work. However, the presentation could be improved and expanded and the link to past work in this area should be more clearly presented in the main publication.

Major points to consider:

Citations to past work are under-done in main text. An example is the work of Toom-Sauntry and Barrie (2002), which described a multi-year weekly snow sampling program from Alert, which is quite similar to this work (with average of 4 day sampling) and should certainly be mentioned in the introduction, but is not. Work at ice core sites (e.g. Summit Greenland, Antarctica) also should be mentioned even if not at the time resolution of this study.

The analysis in this manuscript is presented mostly as a deposition velocity, the ratio of two measured quantities (e.g. Figs. 2, 4, and 5), but the individual parts of this ratio (numerator and denominator) are not discussed as fully. Because both the numerator and denominator go through annual cycles, it is useful to include discussion of both quantities. Specifically, the fall into winter aerosol increase and spring into summer transition is a time of drastic decrease in aerosol loading (e.g. March-May 2015 in Supplemental Table S8), which causes an increase in deposition velocity through equation (1). It would be useful to present graphically in the main text the cycles shown in Table S8 in reference to the snow measurements in Fig. 1. In this way, the scavenging ratio can be better understood. There is also a complementarity between high scavenging and low atmospheric aerosol abundance which can be seen in the data and should be discussed. For example, an very crude order of magnitude of the aerosol lifetime can be estimated by the ratio of the boundary layer height divided by deposition velocity. For sulfate in winter, with v_dep = 0.08, and crudely estimated 1 km boundary layer height, this gives 14 days, and it shortens to 5 days for the "edge seasons" (S/O/My).

Although the method for sampling is unique and useful, it causes "dry deposition" to be a bit different than it might be in true snowpack. This is briefly discussed, but should be more clear. Specifically, as air pumps through actual snowpack, it is filtered by the snow. Harder et al. (1996) indicate that filtering is a significant mechanism of "dry deposition". However, the removal (prior sampling) of old snowpack in the present study causes wind pumping / filtering to not be operative. Thus, the "dry deposition" measured here may be an under-representation of true non-precipitation-related deposition to snowpack. Also, how well particles that dry deposited to the table before snowfall are sampled by scraping off subsequent snowfall is uncertain.

Harder, S. L., S. G. Warren, R. J. Charlson, and D. S. Covert (1996), Filtering of air through snow as a mechanism for aerosol deposition to the Antarctic ice sheet, J. Geophys. Res., 101(D13), 18729–18743, doi:10.1029/96JD01174.

Another related point is that some fluxes with snow are bi-directional (e.g. snow can re-emit species such as ammonia/ammonium, nitric acid/nitrate, bromide, and mercury. In this study, old snow is removed, so emissions from snow are not measurable (and all deposition fluxes are necessarily positive). This could affect interpretation of ammonium, nitrate, and bromide, and should be mentioned for mercury even through their analysis methods did not quantify it. These processes happen in the actual snowpack, and exclusion of consideration of snowpack emissions would make the deposition fluxes measured here overestimates of the net flux for these species.

Specific Comments:

page 1, line 17: The first instance of the word "which" should be replaced by "that" to read "...deposition velocities that encompass all..."

page 3, line 22: Although loss of snow from the table before sampling is mentioned later, it also needs to be discussed here.

page 4, line 10: This reference to a thesis (which is not peer reviewed and may not be always accessible) should be replaced by a brief discussion in the supplemental material. There should not be references to theses in general.

page 6, line 9: The actual quantification of the trajectories should be described here (that is a component of methodology) instead of later.

page 6, line 17: It is also not clear if analytes deposited to a bare table would be effectively sampled from the table into the bottles, so "early dry deposition" could have a number of uncertainties.

page 6, line 19: Are these four occasions of "resuspended" snow the cause of missing data in mid winter discussed later? Make more clear if these refer to the same events.

[Figure]

page 7, line 2: By "free" form, do you mean "dissociated ionic" form? It would also be useful to give some discussion of whether soluble or insoluble species dominated for various class of metals.

Table 1. At some point, method detection limits should be discussed, as well as blanks. For example, the use of nitric acid to clean bottles could lead to a high nitrate blank.

page 9, line 16: This unpublished work is not cited properly. Additionally there should be mention of findings that aerosol (particles and gases) vertical distribution are layered in the Arctic (e.g. Brock et al., Atmos. Chem. Phys., 11, 2423–2453, 2011), and older "Arctic Haze" literature.

Figure 2. I'm confused about what is shown here. It appears that he minimum (presumably the actual lowest value, while prior table showed 25th percentile), median, maximum (similar question). If that is the case, I'm not sure what the "error bars" are. The error bars also seem to be added at the edges of the "box"? Overall, this looks like a "box and whiskers" plot, but it doesn't seem to have the same information as a standard box and whiskers. Please explain further.

Figure 4. In panel a, the "normalized effective deposition velocity" is shown. I'm not sure what that is. The text description doesn't really help much. Is the idea that aerosol particle components are co-deposited and that aerosol particles are internally mixed? Wouldn't I expect all aerosol particle components to then have the same deposition velocity? Figure S3 shows that different components seem to have different average deposition velocities, which could be an indication of external mixture of aerosol components, as was discussed for BC. Some further discussion, and potentially elevation of Figure S3 (or some subset of the species) to the main text would be preferable to the "lumping" that was done in Figure 4, panel a.

Figure 5. I still don't understand the error bars. Please explain.

---

## Author Comment (AC1) · 27 Feb 2017

Referee comments received and published: 6 December 2016 (quoted below)

We thank Referee #1 for their well-thought comments and constructive criticism. The following section recapitulates the referee's comments and provides the authors' response to each point presented by the referee.

Response to Referee Discussion

Referee Comment: This manuscript presents results from a 9-month long campaign that collected samples of freshly fallen snow as soon as possible after each deposition event that occurred at Alert from mid-September 2014 through mid-May 2015. These samples were obtained using a technique that the authors assert allowed estimation of

combined wet and dry deposition over the interval between each collection, and were analyzed for BC, major ions, and a large number of trace elements. The authors also use measurements of the concentration of aerosol-associated BC, ions and a smaller number of trace elements made as part of the baseline suite by station staff to explore the transfer of impurities from the atmosphere to the snow.

Response: We thank the reviewer for this summary.

Referee Comment: The data set is quite rich and interesting, however I feel that the interpretation and synthesis does not exploit the data set well. A major problem is the decision to define and rely on the "effective deposition velocity" (EDV) for nearly all of the discussion. The authors acknowledge on page 2 that wet deposition is dependent on a wide range of factors, many of which change from day to day, such that previous theories have never found it useful to consider a wet deposition velocity. Then on page 9 they point out additional reasons that such a construct is bound to obscure critical details relating to the processes removing aerosol associated (and gas phase) impurities from the atmosphere to the snow surface, without presenting a strong argument why ignoring these details is reasonable.

Response: We agree with the reviewer that an "effective deposition velocity" lumps together both dry and wet deposition, and that this term is not commonly used in the literature. We made this point clearly in the original paper, as pointed out by the reviewer. The reason we chose this approach was simply that it is not possible to accurately separate dry and wet deposition via snow collection/analysis of the type performed in this study. However, discussion of this parameter does provide insight on bulk deposition mechanisms and magnitudes, and indicate when these exceed that expected due to dry deposition alone. Thus, the effective deposition velocity provides a holistic view of how deposition processes change over the winter seasons. As well, the "effective deposition velocity" is a quantity that can be readily extracted from GCM or CTM models, for comparison to the data. Section 3.2.1 has been expanded to give greater clarity on the interpretation and use of this parameter. We thank the reviewer for this comment.

Referee Comment: To my mind this bad decision is compounded by focusing on monthly values of the estimated EDV as the highest temporal resolution (but only shown in Figure S3). In the actual manuscript one of the key figures (Fig. 2) shows estimated EDV for BC and ions, but only as box and whisker plots summarizing the full campaign (minus Jan and Feb which were discarded due to concerns about data quality, more on that later). Another key figure (Fig. 4) shows monthly EDV, but lumps all of the different impurities together, obscuring significant differences that are evident in Fig. S3 with no explanation. Likewise, Fig. 5 does include separate EDVs for BC and the ions, but presents statistical summaries for a 3 month warm season compared to 4 month cold season (again, omitting Jan and Feb).

Response: We agree with the referee that higher resolution effective deposition velocities should be included to allow inferences of specific events. However, we believe monthly values allow better insight into the general deposition regime of Arctic aerosol without the complexity of event-to-event variation. The supplemental Figure S3 has been moved to the manuscript as per the suggestion of both referees, now Figure 2. Six-day values have been added to this figure along with their associated uncertainties. Monthly values are still included for the discussion of variance in deposition by composition and temporally. Section 3.2.2 of the manuscript has been revised as follows to address this issue (Page/line in the revised text: 10/12-23):

Revised text: Monthly effective deposition velocities were used to contrast deposition mechanisms by aerosol composition. A monthly resolution provides insight into the general deposition regime of each analyte, highlighting the impact of bulk deposition characteristics rather than event-specific variability. The variability between aerosol of different composition and the influences of seasonal changes within the Arctic system are simpler to identify without the interference of variability between event-specific conditions. A monthly analysis also facilitates future comparison with modelled results which may not replicate individual events. January and February, 2015, were excluded from the monthly analysis because blizzard and high wind conditions were believed

to have caused significant losses of snow from the snow tables during these months (based on operator reports), which would lead to underestimation of these snow flux values. The effective deposition velocity is best suited to analysis across periods of equal length and precipitation volume, since both of these parameters are inherently included when the wet deposition efficiency is converted to an equivalent deposition velocity. With the exception of January and February, the total monthly snow precipitation over the campaign was relatively constant, with a relative standard deviation of 20%.

Referee Comment: The authors should establish a stronger case for the utility of the EVD, and it should be calculated and presented at the 6-8 day resolution of the aerosol sampling. A revised version of Fig S3 (with approximately 4 times as many data points) should be in the main text, informing most or all of the discussion. In the detailed comments below I will work through the manuscript page by page with a mixture of editorial and content comments. The latter will be based on statements in this draft in relation to the figures that are now included, hence may be less relevant if a revised manuscript focuses on the higher resolution that the data set could provide.

Response: Please see our comments above.

Referee Comment: Before turning to detailed comments I need to raise one other general objection to this draft. The authors claim repeatedly that this data set is unique by virtue of the large number of impurities quantified, that their sampling allowed fluxes to be determined, and that sampling captured all (and only) fresh snow events. I would like to point out that the DGASP campaign at Dye 3 in south Greenland took a very similar approach back in 1988-89, though many of the snow samples were not collected in a way that allowed net accumulation (hence flux) to be quantified. This shortcoming of DGASP sampling was addressed in a long series of summer time campaigns at Summit, Greenland (1989 to at least 2003) that included daily sampling to quantify both concentration and the mass of impurity/unit area in the surface layer of snow (see series of papers by Bergin et al. from the mid 90's with "Flux" in their titles as

examples), and was maintained for year round campaigns at Summit in 1997-1998 and 2000-2002 (Dibb et al., 2007). With better hindsight, the authors may want to rephrase the statements in this draft made on page 1, lines 20-21; page 2, lines 1-2; page 3, lines 2-3; page 3, lines 20-22; page 6, line 12; page 7, lines 3-6 (particularly this one!); page 16, lines 3-5.

Response: Referee #1 referred to two studies which provided similar insight to Arctic snow deposition and should be discussed: the Dye 3 gas and aerosol sampling program (DGASP), which collected atmospheric and snow samples in Greenland between August 1988 and July 1989 (Davidson et al. 1993; Jaffrezo, 1993); and the snow flux papers published by Bergin et al. in the mid 1990's, which consisted of two month-long campaigns collecting snow, fog, and dry deposition in Greenland in the summers of 1992 and 1993 (Bergin et al., 1994; Bergin et al., 1995).

The DGASP study was an excellent campaign with interesting results. Specifically, surface snow samples were collected at Dye 3 following each snow fall during the 1988-89 season, conditions allowing. Samples were analyzed for major ions, metals, optical carbon, and particle size distribution (Jaffrezo, 1993). As mentioned by Referee #1, the deposition area was not measured for all DGASP samples; thus, snow flux could not be calculated. Another distinction between the DGASP campaign and this study is the collection strategy: surface snow collection as compared to snow table collection as per this study. Collection of snow samples from the ground introduces the inherent difficulty of determining the appropriate collection depth and distinguishing fresh and aged snow. Snow table collection avoids this concern; however, both techniques are subject to uncertainty due to redistribution of snow by winds. Thus, the deposition area and time period associated with surface samples may be less certain than that those associated with snow table samples. Furthermore, the collection of dry deposition from between snowfalls would likely differ between the techniques. As mentioned in the paper, it is uncertain what portion of this dry deposition would have been collected from the snow table; however, we believe that this collection efficiency would be similar between

samples. The collection of such dry deposited material in surface snow samples is even harder to quantify, but would be very dependent on small deviations in the collection depth. Thus, we believe the snow table collection technique distinguishes this paper in its ability to quantify the bulk deposition of chemical species as a flux.

The Bergin et al. papers discussed by the referee are also wonderful studies with valuable insight on deposition within the Arctic. These studies quantified daily snow, fog, and dry deposition to collection surfaces at Summit, Greenland during the summers of 1992 and 1993. Samples were analysed for major ions and particle size. The known collection area and time associated with these samples allowed calculation of the material flux and deposition velocity. The Bergin measurements make an excellent complement to the data of this study as they focus on different seasons (summer as compared to fall/winter/spring). Although the measurements of this study are less appropriate for distinguishing the exact split between deposition mechanisms, they do represent a longer collection campaign and broader suite of analytes. As discussed by Bergin et al. (1995), long-term studies are valuable for accessing the annual inventory of deposition within the Arctic.

References to the DGASP and Bergin et al. studies have been added within the introduction, methodology, and results sections (see specific instances of note below). We apologize for not having these in the paper from the outset.

Bergin, M. H., Jaffrezo, J. L., Davidson, C. I., Caldow, R., and Dibb, J.: Fluxes of chemical species to the Greenland ice sheet at Summit by fog and dry deposition, Geochim. Cosmochim. Ac., 58 (15), 3207–3215, doi:10.1016/0016-7037(94)90048-5, 1994.

Bergin, M. H., Jaffrezo, J., Davidson, C. I., Dibb, J. E., Hillamo, R., Maenhaut, M., Kuhns, H. D., and Makela, T.: The Contributions of snow, fog, and dry deposition to the summer flux of anions and cations at Summit, Greenland, J. Geophys. Res., 100 (D8), 16275–16288, doi:10.1029/95JD01267, 1995.

Davidson, C.I., Jaffrezo, J. L., Mosher, B.W., Dibb, J.E., Borys, R.D., Bodhaine, B. A., Rasmussen, R. A., et al.: Chemical constituents in the air and snow at Dye 3, Greenland — I. Seasonal Variations, Atmos. Environ. A-Gen., 27 (17), 2709–2722, doi:10.1016/0960-1686(93)90304-H, 1993.

Jaffrezo, J. L. and Davidson, C. I.: The Dye 3 gas and aerosol sampling program (DGASP): An Overview, Atmos. Environ. A-Gen., 27 (17), 2703–2707, doi:10.1016/0960-1686(93)90303-G, 1993.

Specific lines discussed by the referee are listed below with the revised text by Page/Line(s) #.

1/20-21

[revised manuscript text omitted]

Revised Line: We have left this line as is. Although the referee has provided some additional references, we still find such campaigns to be uncommon. No similar campaign has been completed in the recent decade.

Response to Detailed Comments

Referenced to Page/Line #(s) in the original manuscript:

Referee Comment: as noted above "effective deposition velocities" not established, and maybe not useful, a strong case needs to be made later that the concept does have utility, or these calculated values should not be highlighted in the abstract

Original Line: Comparison with simultaneous measurements of atmospheric aerosol mass loadings yields effective deposition velocities that encompass all processes by which the atmospheric species are transferred to the snow.

Response: As mentioned above, the analysis of effective deposition velocity provides insight into the general deposition regime of the measured analytes which guides the discussion of this paper. Furthermore, this quantity can be directly compared to GCM and CTM output. Because such models cannot often model individual synoptic events, a monthly averaged value is likely of more utility than individual values alone. That said, the individual measurements are now also included in the paper.

Referee Comment: dependent on snow type

Response: Editorial comment corrected in text (revised manuscript page/line 2/28).

2/27-29

Referee Comment: not sure what the message of sentence starting "The efficiencies..." is supposed to be Original Line: The efficiencies of below-cloud scavenging of gases and particles can be similar at an order of magnitude level, though measurements of gaseous scavenging in particular are infrequent and dependent on composition (McMahon and Denison, 1979).

Response: This line has been removed from the revised manuscript.

Referee Comment: Last sentence is quite true, and for pretty good reasons. Why are you bucking the trend?

Original Line: Hence, wet deposition is not typically described via a deposition velocity. Response: As discussed above, the methodology of this study did not allow for accurate split of dry and wet deposition. Thus, a bulk parameter has been presented which allows us to explore the general deposition regime of Arctic aerosols by composition and season. This bulk parameter does provide insight into the degree to which the aerosol transport into snow exceeded that expected due to dry deposition alone.

Referee Comment: Based on the picture in Fig S2 I strongly suspect that the snow table had a major problem with undercatch if there was any wind at all (not just high winds causing blowing snow). It would have been prudent to check this fairly frequently through the campaign by determining the cumulative accumulation of snow on the ground near the table. At a minimum, determination of SWE from either a small pit or a simple core could have been compared to a summation of the SWE for all of the layers collected off the table. Better would have been to check the inventory (mass/unit area) from snow surface to the ground of selected impurities at some interval (at least

monthly), by collecting and analyzing the full column of accumulated snow. In the absence of this kind of validation, you probably need to increase the uncertainty estimate for your impurity fluxes.

Response: These are good points that have now been raised as uncertainties in the paper, in section 2.1.

Referee Comment: What were the QC runs?

Response: Quality control runs were completed by re-analyzing the calibration solution every set number of samples to quantify any instrument drift. An extended supplemental has been prepared with additional details on the methodology. This includes a description of the QC runs for each analysis in supplemental section S2.2.

Referee Comment: How, exactly, was "mass recovery precision" determined? Sounds like some kind of yield tracer was added, but no details given.

Response: Mass recovery was quantified by massing the total sample at several stages over the sample preparation and analysis. This was done to confirm no sample was lost throughout handling. An extended supplemental has been prepared with additional details on the methodology, including the following text: "An average sample mass closure of $\pm 1\%$ was observed over the digestion procedure (i.e., sample mass was measured at all stages of analysis to confirm significant sample was not lost during handling)." (supplemental section S2.1.3).

5/4-6

Referee Comment: When was this "internal standard" added? Before or after digestion? Did it also serve as the yield tracer to track recovery?

Response: The internal standard was added to the digested solution during analysis.

An extended supplemental has been prepared with additional details on the methodology, including the following text: "An internal standard was included in the ICP-MS analysis to quantify and correct for any instrument drift or inter-sample variability. The SCP Science Int. Std. Mix 1 was selected so as to minimize interference with measured analytes while covering the full analyzed spectrum of mass to charge ratios. This internal standard was added to the digested solution as they were sampled for analysis." (supplemental section S2.2)

Referee Comment: in a class 100

Response: Editorial comment corrected in text (5/10).

5/25-26

Referee Comment: Are U. Toronto MA thesis publicly available? If not, are they citable in ACP?

Response: Pertinent details of this thesis have been moved into the supplemental, section S2, and the reference removed.

6/14-24

Referee Comment: There is a sizeable literature about the problems with surrogate surfaces for the collection of both dry and wet deposition. Some of that should be cited here.

Response: We agree that additional references should be provided here and apologise for not doing so. The following have been added (6/18-19):

Davidson, C. I., Lindberg, S. E., Schmidt, J. A., Cartwright, L. G., and Landis, L. R.: Dry deposition of sulfate onto surrogate surfaces, J. Geophys. Res., 90 (D1), 2123–30, doi:10.1029/JD090iD01p02123, 1985.

[Figure]

Hicks, B. B.: Measuring dry deposition: A Re-assessment of the state of the art, Water Air Soil Poll., 30, 75–90, doi:10.1007/BF00305177, 1986.

Ibrahim, M., Barrie, L. A., and Fanaki, F.: An Experimental and theoretical investigation of the dry deposition of particles to snow, pine trees and artificial collectors, Atmos. Env., 17 (4), 781–88, doi:10.1016/0004-6981(83)90427-4. 1983.

6/24-25

Referee Comment: May want to include comparison between cumulative depth, or cumulative SWE probably better, on the table and at the met station. This would allow the reader some insight into how bad table undercatch may have been (or persuade you to make some correction).

Response: While we agree that the depth of snow captured by the snow tables was likely impacted by winds, we do not believe the local meteorological station provides a better estimate of the actual precipitation depth. The Alert ECCC meteorological stations are over 6 km away with a 50 m difference in elevation. Such a difference in location could lead to significant differences in observed snowfall. Furthermore, observations of the collection station operators showed many instances of disagreement with the meteorological station record. For example, there were periods when the operators noted snowfall and yet the meteorological station saw none, or vice-versa. There were also times when the operators noted blowing snow, yet the meteorological station did not indicate this. Thus, we do not believe that the snow depth measured by the station would be an appropriate comparison to that seen at the collection site. Please see revised line of 3/20-22 above.

However, we have performed a supplementary analysis with the assumption that the snow depths measured at the meteorological station are applicable to the snow collection site. These adjusted median effective deposition velocities ranged from 0.08 to 1.7 cm/s, as compared to 0.03 to 1.1 cm/s using snow table depths. The relative velocities and interpretation of the deposition characteristics remained unchanged.

Referee Comment: Flux per day over 2-18 days makes no sense for a wet deposition event likely lasting just hours. Understood that you cannot separate wet and dry in this data set, but this contributes to discomfort with the calculated EVDs.

Response: The referee is quite right that the deposition measured does not relate to that of a single snowfall but total deposition over a period. As per the discussion above about the utility of the effective deposition velocity, total flux over a period provides insight into the bulk movement of material in the Arctic atmosphere.

8 (Fig 1)

Referee Comment: Why are there gaps of different lengths and at different times in the time series of BC, ions, and crustal metals? According to section 2.1, anytime one bottle was filled there should have been 3 others (presumably for the different analyses). Problems causing gaps need to be noted and enumerated (probably better in 2.1 than here).

Response: There were events when insufficient snow volume fell to fill all sample bottles. In these cases a sub-set of the samples was collected. The samples collected on each date are listed in Table S1. The following text has been added to section 2.1 "When insufficient snow volume was available for complete collection, a subset of the replicate samples was collected as listed in supplemental Table S1." (3/20-21).

9/13-16

Referee Comment: Why are the much more numerous flights from the 4 NOAA AGASP campaigns or the 5-6 NSF TOPSE flights not included here? My impression is that some of these showed significant gradients in the lowest 3 km. (Note that the special issue of Atmos Environ on DGASP also includes AGASP3 papers and provides access to the earlier ones.)

Response: Although some previous studies have shown the lower Arctic atmosphere

to be somewhat more consistent, upon further investigation we agree with the referee that this is not always the case. Therefore, we have rephrased this section as follows to discuss this limitation: "A caveat to this analysis is that the three deposition mechanisms relate to different atmospheric concentrations, a gradient which is not necessarily captured when the ground-level atmospheric concentration (CA) is used to calculate the effective deposition velocity: dry deposition affects the lower atmosphere, in-cloud scavenging the cloud layer, and below-cloud scavenging the full below-cloud atmospheric column. Previous observations of vertical profiles in the Arctic have shown notable variability with altitude (Hansen and Rosen, 1984; Leaitch et al., 1989; Spackman et al., 2010; Brock et al., 2011; Sharma et al., 2013). So, the calculated effective velocity includes an intrinsic variability dependent on the vertical atmospheric profile of each analyte." (9/30-10/4).

Brock C. A., Cozic, J., Bahreini, R., Froyd, K. D., Middlebrook, A. M., McComiskey, A., Brioude, J., et al.: Characteristics, sources, and transport of aerosols measured in spring 2008 during the aerosol, radiation, and cloud processes affecting Arctic Climate (ARCPAC) Project; Atmos. Chem. Phys., 11, 2423–2453, doi:10.5194/acp-11-2423-2011, 2011.

Hansen, A. D. A., and Rosen, H.: Vertical distributions of particulate carbon, sulfur, and bromine in the Arctic haze and comparison with ground-level measurements at Barrow, Alaska, Geophys. Res. Lett., 11 (5), 381–84, doi:10.1029/GL011i005p00381, 1984.

Sharma, S., Ishizawa, M., Chan, D., Lavoué, D., Andrews, E., Eleftheriadis, K., and Maksyutov, S.: 16-Year simulation of Arctic black carbon: Transport, source contribution, and sensitivity analysis on deposition, J. Geophys. Res-Atmos., 118, 943–964, doi:10.1029/2012JD017774, 2013.

Leaitch, W. R., Hoff, R. M., and MacPherson, J. I.: Airborne and lidar measurements of aerosol and cloud particles in the troposphere over Alert Canada in April 1986, J. Atmos. Chem., 9, 187–211, doi:10.1007/BF00052832, 1989.

Spackman, J. R., Gao, R. S., Neff, W. D., Schwarz, J. P., Watts, L. A., Fahey, D. W., Holloway, J. S., et al.: Aircraft observations of enhancement and depletion of black carbon mass in the springtime Arctic, Atmos. Chem. Phys., 10, 9667–9680, doi:10.5194/acp-10-9667-2010, 2010.

9/16-18

Referee Comment: Somewhat bad form to use unpublished data to support your position, but beyond that, flights in April are not really informative about stability and stratification in the winter. Recall, you are asserting these data are unique because they include the cold season (see later comments related to your Figs 4 and 5).

Response: We agree with the referee that this is a very good point. The reference to unpublished data has been removed. The discussion of vertical variability has been revised as described in the previous point.

9/21-27

Referee Comment: As noted, assuming you can make a better case for utility of EVDs, I do not like the use of monthly averages here and do not think the final sentence is valid justification. There are a lot of papers focused on the POLARCAT airborne campaigns where global and regional CTMs attempted to simulate day by day, plume by plume in some cases, and more should be expected as part of recent and future campaigns. Modelers are striving to improve their simulations, so if experimentalists provide high resolution records they welcome the challenge of simulating them, however poorly the first round of comparisons turn out.

Response: Yes, we agree that some models are capable of high-resolution results; however, monthly values would be of use to a variety of lower-resolution models and seasonal trends can be compared across models simulating different years. Please refer to response above discussing the use of monthly values within the paper's discussion.

Referee Comment: As noted earlier, summarizing 7 months in box and whisker plot makes little sense to me.

Response: Please see response above.

10/10-13

Referee Comment: To me, the most puzzling aspect of Fig 2 is why Ca is so much higher than Na, Cl and Mg. This is not even mentioned until page 15 (with no plausible explanation). Similarly, suggesting here that BC, NH4 and SO4 seem dominated by dry dep causes some tension with later suggestion that they are serving as CCN (or at least being efficiently scavenged) in S/O/My (Fig 5).

Response: Discussion of the high Ca velocity has been moved to this section (13/15-17). As mentioned in the original manuscript, we do not have sufficient information to make a definite claim as to the processes resulting in the elevated Ca deposition relative to Mg. However, the previous study quoted in the text (Banta et al., 2008) showed that distinct Ca-rich and Mg-rich crustal particles may typically exist within the Arctic. Therefore, we suggest that the discrepancy is a result of differing properties associated with these different crustal sources.

We show that the effective deposition velocities of BC, NH4+, and SO42- over the colder months fall within the typical range of dry deposition velocities for accumulation mode particles. This indicates that dry deposition is a dominant deposition mechanism for these chemical species over cold months. Within the warmer months, the effective deposition velocities of BC, NH4+, and SO42- are still relatively low but do show some enhancement above the typical dry deposition range. Thus, we suggest that they experience enhanced contribution from wet deposition over this period. Specifically, this enhance wet deposition may be related to their activity as CCN. However, dry deposition is likely still a significant factor for these species over warm months as well. We do

not believe these statements are contradictive.

11/5-6

Referee Comment: Not so sure this is true. The Hi-Vol sampling at Alert used to use cellulose filters (Whatman 41s) which do collect acids with decent efficiency. And for a filter exposed for 6-8 days, seasalt and dust can build up to the point where acids may be collected quantitatively.

Response: The Hi-Vol collection was completed using cellulose Whatman 41 filters, as has been the procedure at Alert for decades (Barrie and Hoff, 1985). We agree that although these filters are intended to sample particles they are subject to both positive and negative artefacts. Gases may adsorb onto the filter or acidic gases may react with alkaline sea salt or dust, if they are present, and thus become part of the sample. Gases may also re-volatilize due the high sample flow, or particles may react with acidic Sulphur species on the filter and be released as a gas. However, given the observed chloride deficit in the Hi-Vol aerosol sample, we suspect that the collection of gaseous species by Hi-Vol sampling was likely not a major contribution.

The text has been revised to clarify that Hi-Vol measurements are likely dominated by particle collection, though not necessarily absent of gaseous aerosol collection. Original Line: Due to the nature of the Hi-Vol analysis, the measured atmospheric concentrations include only the particle-phase ambient species; however, snow can scavenge both the particle and gas phases, and thus snow samples provide a composite measurement of both. Revised Line: It is expected that the Hi-Vol measurement technique would collect predominantly atmospheric particles, while snow would scavenge both gaseous and particulate aerosol (12/17-18). Barrie, L. A., and Hoff, R.M.: Five years of air chemistry observations in the Canadian Arctic, Atmos. Environ., 19 (12), 1995–2010, doi:10.1016/0004-6981(85)90108-8, 1985.

10/7 to 11/8

Referee Comment: Chloride deficit in MBL aerosol and excess in Arctic snow (especially in summer) has been well documented by many others in addition to Toom-Sauntry and Barrie, 2002. This section seems too long for significance of the point being made, but also lacking references to prior work.

Response: We agree that this section need not include such a detailed discussion of chloride deficit. The bulk of this discussion and associated plot have been moved to the supplemental. Furthermore, additional references have been added (12/11):

Barrie, L. A., and Hoff, R.M.: Five years of air chemistry observations in the Canadian Arctic, Atmos. Environ., 19 (12), 1995–2010, doi:10.1016/0004-6981(85)90108-8, 1985.

Quinn, P. K., Bates, T. S., Schulz, K., and Shaw, G. E.: Decadal trends in aerosol chemical composition at Barrow, AK: 1976–2008, Atmos. Chem. Phys., 9, 18727–43, doi:10.5194/acpd-9-18727-2009, 2009.

Referee Comment: enhanced deposition

Response: Editorial comment corrected in text (13/12).

12/18-21

Referee Comment: As pointed out in first sentence of this paragraph, everything shown in Fig 2 has higher EVD than BC, so what is so special about these 4? They are not as enhanced as Ca in the median, and max EVD for K, Mg, C2O4 and Ca are all higher than the max for MSA, Br and Cl. I agree that the 4 you highlight can have appreciable gas phase, but that is not something discovered or even really supported by Fig 2.

Original Line: Gas-phase deposition to snow is suggested to contribute to the observed enhanced velocities of methanesulphonate (MSA), Br, Cl, and NO3 relative to BC, either due to gas-phase emissions or gas-phase partitioning during aerosol aging

followed by subsequent deposition.

Response: As stated by the referee these four analytes were specified as they can all have appreciable gas phase. We do not claim to have discovered this point, and believe the text reflects this. Rather we have simply shown that our observations are consistent with this and that the gas-phase deposition can be an important factor in the bulk deposition of chemical species within the Arctic.

Pages 12-14 (Figs 4 and 5)

Referee Comment: As noted at top, I like that 4 at least shows monthly resolution, but don't like lumping all impurities, while 5 is nice in showing BC and all ions, but is averaging over too much time.

More importantly, I think both are missing key point (which calls the value of EVD into question). Perhaps the most important feature of the Arctic is the huge seasonal contrast in daylight, which leads to big contrasts in energy balance and atmospheric stability. In the winter the atmosphere is extremely stable with weak vertical mixing. Atmospheric impurities that get into the BL tend to be trapped there, and concentrations can be high. As things warm up the BL deepens and concentrations tend to decrease close to the surface. This is clearly reflected in Table S8 which shows very low concentrations of BC, Cl, C2O4, Na, K, Mg in SOMy compared to the colder months. Pattern is not so strong for NO3, NH4 and SO4, but the averages for the 3 warm months are quite low compared to the 6 (or 4) cold ones. Part of this may be transport related, and local biogenic emissions may account for the weaker trend in SO4 and opposite trend in MSA, but dilution into a much deeper BL is probably an important factor.

As a thought experiment, given constant monthly accumulation of snow, and constant burden of impurities in the BL, and assuming complete cleansing of the BL over each month, but a 5-10 fold deeper and well mixed BL in warm months compared to cold ones, what would the EVDs look like. Constant monthly flux to snow divided by 5 (or 10) fold lower aerosol concentration measured at surface would yield 5 (or 10) times larger

[Figure]

EVD in the warm months. Clearly, the story is not so simple. But, is it really plausible that scavenging in mixed phase clouds is the major factor accounting for increase in BC and all the ions (which should all behave differently in cloud)?

See again the suggestion to include a higher resolution version Fig S3 as the focus of discussion, and exploit the differences to gain insight.

Response: The focus of our discussion is the seasonal variation in bulk deposition processes. We agree that there is significant and interesting variability in deposition at an event resolution; however, the influences on individual events are very complex and cannot be accurately judged without additional information. A holistic view of the scavenging and deposition process can be obtained through analysis and discussion of monthly values to provide insight into their gross mechanisms and better differentiate seasonal trends from the random variability between snowfall events. We have added high-resolution data to the manuscript to guide discussion and future research, but have kept seasonal variation as the focus of our analysis.

We agree that the boundary layer height could be an influence on deposition, as stated by the referee. We have obtained radiosonde measurements from Alert to better characterize the vertical profile. This added analysis is described in section 2.2 in the revised manuscript. Briefly, temperature and relative humidity profiles were used to estimate the mixing height and cloud height over the collection campaign. Both these parameters were considered as potential influences on the deposition process: mixing height essential as a dilution factor on the ambient concentration, and cloud height as a controlling factor on the height of the atmospheric column scavenged by below-cloud wet deposition. These characteristics have been added to the revised Figure 4 and compared to the observed trends in deposition velocity. While the heights do show moderate correlation with the effective deposition velocities, their correlation is lower than that of temperature. Furthermore, the heights are observed to be primarily episodic, unlike the distinct seasonal exhibited by temperature. Hence, the hypothesis that cloud phase may be an important factor in seasonal changes in deposition

processes over the cold season was maintained. Furthermore, the radiosonde data showed temperatures above -20 °C to be common within the cloud layer in the warmer months. This supports the potential presence of mixed-phase clouds during this time.

Please also note the supplement to this comment:
http://www.atmos-chem-phys-discuss.net/acp-2016-944/acp-2016-944-AC1-supplement.pdf

---

## Author Comment (AC2) · 27 Feb 2017

Referee comments received and published: 6 January 2017 (quoted below)

We thank Referee #2 for their many excellent suggestions and comments. The following section recapitulates the referee's comments and provides the authors' response to each point presented by the referee.

Response to Referee Discussion

Referee Comment: This manuscript describes measurements of the chemical composition of freshly precipitated snow from Alert, Canada and relates those snow measurements to aerosol measurements to derive deposition velocities for various species. The manuscript reads well, and describes a unique contribution to the literature and

thus is appropriate, after revision for eventual publication in ACP. The work also uses a comprehensive set of multi-species analyses (BC, IC, ICP-MS) that are a valuable expansion on prior work. However, the presentation could be improved and expanded and the link to past work in this area should be more clearly presented in the main publication.

Response: We thank the referee for their comments.

Referee Comment: Citations to past work are under-done in main text. An example is the work of Toom-Sauntry and Barrie (2002), which described a multi-year weekly snow sampling program from Alert, which is quite similar to this work (with average of 4 day sampling) and should certainly be mentioned in the introduction, but is not. Work at ice core sites (e.g. Summit Greenland, Antarctica) also should be mentioned even if not at the time resolution of this study.

Response: We agree with the referee and have added these references to the papers (Page/line within the revised text: 2/2-5).

Referee Comment: The analysis in this manuscript is presented mostly as a deposition velocity, the ratio of two measured quantities (e.g. Figs. 2, 4, and 5), but the individual parts of this ratio (numerator and denominator) are not discussed as fully. Because both the numerator and denominator go through annual cycles, it is useful to include discussion of both quantities. Specifically, the fall into winter aerosol increase and spring into summer transition is a time of drastic decrease in aerosol loading (e.g. March-May 2015 in Supplemental Table S8), which causes an increase in deposition velocity through equation (1). It would be useful to present graphically in the main text the cycles shown in Table S8 in reference to the snow measurements in Fig. 1. In this way, the scavenging ratio can be better understood. There is also a complementarity between high scavenging and low atmospheric aerosol abundance which can be seen in the data and should be discussed. For example, an very crude order of magnitude of the aerosol lifetime can be estimated by the ratio of the boundary layer height divided

by deposition velocity. For sulfate in winter, with v_dep = 0.08, and crudely estimated 1 km boundary layer height, this gives 14 days, and it shortens to 5 days for the "edge seasons" (S/O/My).

Response: We agree that including the individual data, both ambient and snow flux, in the paper is appropriate, along with the derived deposition velocities. The discussion of snow measurements in section 3.1 has been extended. Higher resolution atmospheric measurements have been added to the supplemental, now section S3, and briefly discussed within the text, section 3.1. Although a decrease in atmospheric concentration is related to increased deposition velocity given constant flux, we believe it is important to note that this is a complex process and flux would not necessarily remain constant under reduced atmospheric loads. The deposition velocity is related to the processes controlling deposition, not just the absolute magnitude of the ambient aerosol load.

The thought exercise proposed by the referee is useful as a high level comparison between chemical species. However, the Arctic system is especially complicated by long-range transport and persistent inversion layers which make calculation and interpretation of aerosol lifetimes very difficult. We worry that simplistic pseudo lifetimes as suggested by the referee may be misconstrued by readers and lead to confusion. Therefore, we have not added this discussion to the manuscript.

Referee Comment: Although the method for sampling is unique and useful, it causes "dry deposition" to be a bit different than it might be in true snowpack. This is briefly discussed, but should be more clear. Specifically, as air pumps through actual snowpack, it is filtered by the snow. Harder et al. (1996) indicate that filtering is a significant mechanism of "dry deposition". However, the removal (prior sampling) of old snowpack in the present study causes wind pumping / filtering to not be operative. Thus, the "dry deposition" measured here may be an under-representation of true non-precipitation-related deposition to snowpack. Also, how well particles that dry deposited to the table before snowfall are sampled by scraping off subsequent snowfall is uncertain.

Harder, S. L., S. G. Warren, R. J. Charlson, and D. S. Covert (1996), Filtering of air through snow as a mechanism for aerosol deposition to the Antarctic ice sheet, J. Geophys. Res., 101(D13), 18729–18743, doi:10.1029/96JD01174.

Response: This is a very good point concerning the dry deposition mechanism via air pumping, and this caveat has been added to the paper.

Revised line: Furthermore, it should be noted that dry deposition via the filtration of air as it is pumped through the snowpack (as described in Harder et al., 1996) may differ between snow on a snow table and that on the surface. (4/8-9)

Referee Comment: Another related point is that some fluxes with snow are bi-directional (e.g. snow can re-emit species such as ammonia/ammonium, nitric acid/nitrate, bromide, and mercury. In this study, old snow is removed, so emissions from snow are not measurable (and all deposition fluxes are necessarily positive). This could affect interpretation of ammonium, nitrate, and bromide, and should be mentioned for mercury even through their analysis methods did not quantify it. These processes happen in the actual snowpack, and exclusion of consideration of snowpack emissions would make the deposition fluxes measured here overestimates of the net flux for these species.

Response: This point has been mentioned in the revised version of the paper and we have also revised the text to recognize that re-emission of deposited species from the snow that we do study are still operative, i.e. some bi-directional activity may be occurring during the timescale over which the snow sits on the snow table.

Revised Line: The collection of fresh snow samples reduces the impact of snow sublimation and/or melt as well as the movement of chemical species between snow and air, which can be a concern for snowpack sampling; however, some bi-directional exchange between snow and atmosphere is unavoidable and natural within natural snowpack systems. (3/28-30)

Response to Detailed Comments

Referenced to Page/Line #(s) in the original manuscript:

Referee Comment: The first instance of the word "which" should be replaced by "that" to read "...deposition velocities that encompass all...”

Response: Editorial comment corrected in text (revised manuscript page/line: 1/19).

Referee Comment: Although loss of snow from the table before sampling is mentioned later, it also needs to be discussed here.

Response: This change has been made (4/1-8).

Referee Comment: This reference to a thesis (which is not peer reviewed and may not be always accessible) should be replaced by a brief discussion in the supplemental material. There should not be references to theses in general.

Response: Pertinent details of this thesis have been moved into the supplemental, section S2, and the reference removed.

Referee Comment: The actual quantification of the trajectories should be described here (that is a component of methodology) instead of later.

Response: We have moved this section earlier (6/9-12).

Referee Comment: It is also not clear if analytes deposited to a bare table would be effectively sampled from the table into the bottles, so "early dry deposition" could have

a number of uncertainties.

Original Line: "Firstly, dry deposition at the beginning of each period would fall directly on the exposed clean table rather than onto previously deposited snow. The difference in surface characteristics created unknown uncertainty in the deposition rate and collection efficiency of the initial portion of dry deposition compared to that which deposited onto a snow-covered table."

Response: This line is meant to communicate that it was uncertain what proportion of the dry deposition to the bare table (i.e., the "initial portion") was collected. To clarify this point, the line has been revised to "Firstly, dry deposition at the beginning of each period would fall directly on the exposed clean table rather than onto previously deposited snow. It is unknown what impact these different surface characteristics could have had on the initial deposition rate and collection efficiency. Thus, there is additional uncertainty in the capture of initial dry deposition to the bare table." (6/19-22).

Referee Comment: Are these four occasions of "resuspended" snow the cause of missing data in mid winter discussed later? Make more clear if these refer to the same events.

Response: Events when blowing snow was observed were excluded from this analysis. Beyond these events, there were additional dates in which some or all samples were missed (typically due to limited snowfall volumes). The following sentence has been added following the line in question to clarify: "The dates of these blowing snow events are noted in Table S1 as are missed collections." (6/25).

Referee Comment: By "free" form, do you mean "dissociated ionic" form? It would also be useful to give some discussion of whether soluble or insoluble species dominated for various class of metals.

Response: Yes, the terminology "free form", as used in this line, is in reference to the "dissociated ionic" form. This wording has been changed as suggested to provide better clarity (7/2). A brief discussion of the soluble and insoluble portions measured for ICP-MS metals has been added in section 3.1: "The metal measurements can be roughly classified into three categories: predominantly insoluble analytes Al and Fe (>50% insoluble over full campaign); variably soluble/insoluble analytes As, Pb, Cu, Ba, Ti, Mn, K, Cd, and Mg; and predominantly soluble analytes Ca, Cr, Co, and Na (<50% insoluble) (in order from least to greatest soluble fraction), excluding analytes with insufficient soluble or insoluble measurements above MDL." (7/3-7).

Table 1

Referee Comment: At some point, method detection limits should be discussed, as well as blanks. For example, the use of nitric acid to clean bottles could lead to a high nitrate blank.

Response: An expanded supplemental has been provided with additional details on the methodology and uncertainty analysis. Supplemental section S2.3 now provides the uncertainty calculation used for measurements along with the calculated error fraction, method detection limit, and signal-to-noise for each analyte in Table S6.

Referee Comment: This unpublished work is not cited properly. Additionally there should be mention of findings that aerosol (particles and gases) vertical distribution are layered in the Arctic (e.g. Brock et al., Atmos. Chem. Phys., 11, 2423–2453, 2011), and older "Arctic Haze" literature.

Response: The unpublished reference has been removed from the text. The text has been revised as follows to incorporate the additional references: "A caveat to this analysis is that the three deposition mechanisms relate to different atmospheric concentrations, a gradient which is not necessarily captured when the ground-level atmospheric

concentration (CA) is used to calculate the effective deposition velocity: dry deposition affects the lower atmosphere, in-cloud scavenging the cloud layer, and below-cloud scavenging the full below-cloud atmospheric column. Previous observations of vertical profiles in the Arctic have shown notable variability with altitude (Hansen and Rosen, 1984; Leaitch et al., 1989; Spackman et al., 2010; Brock et al., 2011; Sharma et al., 2013). So, the calculated effective velocity includes an intrinsic variability dependent on the vertical atmospheric profile of each analyte." (9/30-10/4).

Brock C. A., Cozic, J., Bahreini, R., Froyd, K. D., Middlebrook, A. M., McComiskey, A., Brioude, J., et al.: Characteristics, sources, and transport of aerosols measured in spring 2008 during the aerosol, radiation, and cloud processes affecting Arctic Climate (ARCPAC) Project; Atmos. Chem. Phys., 11, 2423–2453, doi:10.5194/acp-11-2423-2011, 2011.

Hansen, A. D. A., and Rosen, H.: Vertical distributions of particulate carbon, sulfur, and bromine in the Arctic haze and comparison with ground-level measurements at Barrow, Alaska, Geophys. Res. Lett., 11 (5), 381–84, doi:10.1029/GL011i005p00381, 1984.

Sharma, S., Ishizawa, M., Chan, D., Lavoué, D., Andrews, E., Eleftheriadis, K., and Maksyutov, S.: 16-Year simulation of Arctic black carbon: Transport, source contribution, and sensitivity analysis on deposition, J. Geophys. Res-Atmos., 118, 943–964, doi:10.1029/2012JD017774, 2013.

Leaitch, W. R., Hoff, R. M., and MacPherson, J. I.: Airborne and lidar measurements of aerosol and cloud particles in the troposphere over Alert Canada in April 1986, J. Atmos. Chem., 9, 187–211, doi:10.1007/BF00052832, 1989.

Spackman, J. R., Gao, R. S., Neff, W. D., Schwarz, J. P., Watts, L. A., Fahey, D. W., Holloway, J. S., et al.: Aircraft observations of enhancement and depletion of black carbon mass in the springtime Arctic, Atmos. Chem. Phys., 10, 9667–9680, doi:10.5194/acp-10-9667-2010, 2010.

Figure 2

Referee Comment: I'm confused about what is shown here. It appears that he minimum (presumably the actual lowest value, while prior table showed 25th percentile), median, maximum (similar question). If that is the case, I'm not sure what the "error bars" are. The error bars also seem to be added at the edges of the "box"? Overall, this looks like a "box and whiskers" plot, but it doesn't seem to have the same information as a standard box and whiskers. Please explain further.

Response: The original Figures 2 and 5 have been updated to avoid misunderstanding (now Figures 3 and 5). The new plots do not include a bar and whisker design. Instead each monthly value is plotted as a point with the median and full range $\pm$ uncertainty shown.

Figure 4

Referee Comment: In panel a, the "normalized effective deposition velocity" is shown. I'm not sure what that is. The text description doesn't really help much. Is the idea that aerosol particle components are co-deposited and that aerosol particles are internally mixed? Wouldn't I expect all aerosol particle components to then have the same deposition velocity? Figure S3 shows that different components seem to have different average deposition velocities, which could be an indication of external mixture of aerosol components, as was discussed for BC. Some further discussion, and potentially elevation of Figure S3 (or some subset of the species) to the main text would be preferable to the "lumping" that was done in Figure 4, panel a.

Response: The use of normalized effective deposition velocities was simply to allow easier comparison among the analyzed chemical species and to explore a "typical" seasonal pattern. For each analyte the effective deposition velocity for each month was divided by the average effective deposition velocity of that analyte to obtain a normalized value. However, since the supplemental Figure S3 has been moved to the manuscript (now Figure 2), as per the suggestion of both referees, the normalized
trend is no longer required for comparison and has been removed. The provided meteorological trends have now been compared to the trends of individual analytes within the revised section 3.2.3.

Figure 5

Referee Comment: I still don't understand the error bars. Please explain.

Response: Please see response to comment on the original Figure 2.

Please also note the supplement to this comment:
http://www.atmos-chem-phys-discuss.net/acp-2016-944/acp-2016-944-AC2-supplement.pdf

---

## Author Response (AR2)

**Observations of Atmospheric Chemical Deposition to High Arctic Snow – Second Response to Anonymous Referee #1**

Referee comments received and published: 14 Mar 2017 (quoted below in blue text)

We would like to thank Referee #1 for their detailed comments and discussion. We greatly appreciate the care with which the referee has reviewed this manuscript and the improvements gained through their insight.

**Response to Referee Discussion**

Referee Comment: This draft of the manuscript is significantly improved from the original submission. I am still far from convinced that the "effective deposition velocity" is a useful construct, but it is obvious that the authors feel that it is. I also feel that the further attempt to explain the very high EVD for Ca^2+ is not convincing. It is troubling that the detection limit (reported in Supplemental Table 6) is so much higher (more than a factor of 7) than all other ions, suggesting very large variability in the blanks, or a serious analytical problem. (Note this is not true for the aerosol composition detection limits in Supplemental Table 8). The authors assert that qualitative agreement with ICPMS measurements of Ca gives confidence in both data sets, but only about half of all possible samples were above detection limits on both instruments. I find it hard to believe that so many samples were below ICPMS detection limits, raising concern about that data set as well. Given that the authors can not really explain the high EVD, they might be better served leaving Ca^2+ out of the story due to analytical issues causing large uncertainties.

Response: The high method detection limit (MDL) of $Ca^{2+}$, as pointed out by the referee, is a result of variation in the shipping/handling blanks (Hblks). There were five of these blanks collected over the course of the campaign, roughly one per month. These bottles were opened at Alert, but with no snow collected, and then transported and analysed alongside the other samples to capture any background associated with the full handling process. Of these five Hblks, three were below the $Ca^{2+}$ instrument detection limit (IDL) of 18 ppb, and two were above IDL at 50 and 79 ppb. The MDL for each compound was calculated as the maximum of the following:

- IDL
- Three standard deviations of the Hblks
- Three standard deviation of the preparation blanks

Thus, for $Ca^{2+}$, the MDL was set to three standard deviation of the five Hblks. This is a conservative approach since the three Hblks below IDL were very imprecise, but reflects the full variability in the potential shipping/handling background. While all but one $Ca^{2+}$ samples were well above the IDL, only 63% were above the calculated MDL. The other ions measured by IC did not show such a large variability in HBlks, leading to their lower MDL values.

To estimate the impact of this potential background we have used the MDL to get a "background effective deposition velocity". Dividing the MDL by the maximum snow depth and the maximum atmospheric concentration give a conservative background of about 0.45 cm/s (note, this background is over exaggerated since these maxima do not coincide). Subtracting this exaggerated background from the calculated $Ca^{2+}$ effective deposition velocities, the warm month $Ca^{2+}$ median would still be very high at about 2.2 cm/s and the cold month median would be more similar to the other particle-dominated analytes at about 0.4 cm/s. Thus, we believe the observation of high $Ca^{2+}$ deposition rates in the warm months to be reasonable, even if we do not at present have an explanation for the anomaly. Figures 2, 3, and 5 in the manuscript show the associated uncertainty about the $Ca^{2+}$

deposition velocity to intersect zero for several cold months; thus we believe it is clear that the cold-month deposition velocity is not significantly different from those of other analytes.

We have also considered the $Ca^{2+}$ values relative to other crustal-related analytes. Figure 5 shows that the maximum monthly deposition velocities for $K^+$ and $Mg^{2+}$ are similar to or greater than the maximum $Ca^{2+}$ monthly velocity. In fact, if the average velocity is considered instead of the median, $K^+$ has the highest warm-month value at 3.2 cm/s. Furthermore, the ratios of $Ca^{2+}$ to $Mg^{2+}$ in this study was found to be typical compared to those seen by others (as listed in supplemental Table S5: Ross and Granat, 1986; Li and Winchester, 1993; Osada et al., 1996; Toom-Sauntry and Barrie, 2002; Dibb et al., 2007).

While we agree that uncertain measurements should not be included in the analysis, we believe the $Ca^{2+}$ to be sufficiently substantiated to keep in the paper with the provided information on data uncertainty. Although we do not have an explanation for the discrepancy between $Ca^{2+}$ and $Mg^{2+}$ we hope that a reasonable hypothesis may be proposed in the future and perhaps spur future research in this area.

Referee Comment: One thing that the authors may want to expand on a bit more is the suggestion that the study period may have been unusual due to the active Icelandic volcano (mentioned in just one sentence near the end of discussion; pg 17 lines 2-5). Provision of the detailed aerosol data in Supplement Table 7 really makes it clear that the canonical "Arctic Haze" pattern for sulfate (and other pollution and dust derived constituents that usually covary with sulfate) with elevated concentrations in late winter into spring, generally peaking DJF, was not obvious. The really striking observation to me is that sulfate deposition to/with snow is so weak in the winter time, even in a year that may have had the usual haze peak in the atmosphere plus additional enhancement from the volcano. Cliff Davidson suggested very weak scavenging of sulfate by winter snow near Dye 3 way back (several papers before DGASP), based on snowpits that showed winter enhancements that were not proportional to increase in aerosol sulfate at Barrow, Alert, NyAesund during the haze season. DGASP showed that Haze really does not impact sites high on the Greenland ice sheet, so the inferred weak scavenging was not confirmed over Greenland. (Cliff actually claimed marked seasonality of sulfate scavenging, stronger in summer, since he did have aerosol measurements in summer that showed low sulfate compared to what he assumed it was in winter/spring.) Now it may be that you are confirming weak removal of sulfate during winter for Alert, and perhaps other parts of the Arctic basin, but this probably means untangling any impacts on this winter from the volcano.

Response: We agree with the referee that this is an interesting observation. The unusual fall peak in $SO_4^{2-}$ is also mentioned earlier in the text while comparing the observed snow mixing ratios to previous studies (revised page 9 /lines 9-10). To emphasize that the observations of $SO_4^{2-}$ may not be typical of Arctic Haze the following line has been added to the manuscript: "*Thus, the $SO_4^{2-}$ observations of this campaign, especially the peak snow mixing ratio seen in the fall, may not reflect a seasonal trend for typical Arctic Haze.*" (16/5-6)

We believe the Davidson study referred to by the referee is the 1987 study listed below. This Davidson study was referenced in the manuscript in regards to the typical range of dry deposition velocities but not for the temporal variability in deposition. The following line has been added to the manuscript to correct this oversight: "*This aligns with seasonal trend in $SO_4^{2-}$ deposition observed by Davidson et al. (1985b and 1987) and the suggested link to deposition mechanism.*" (14/29-30)

Davidson, C. I., Honrath, R. E.. Kadane, J. B., Tsay, R. S., Mayewski, P. A. Lyons, W. B., and Heidam, N. Z.: The Scavenging of atmospheric sulfate by Arctic snow, Atmos. Environ. A-Gen., 21 (4), 871–882, doi:10.1016/0004-6981(87)90083-7, 1987.

Davidson, C. I, Santhanam , Fortmann, S. R. C., and Marvin, P. O.: Atmospheric transport and deposition of trace elements onto the Greenland ice sheet, Atmos. Environ., 19 (12), 2065-2081, 1985b.

Referee Comment: As noted last time, the authors really need to look at Dibb et al., 2007 (Seasonal variations in the soluble ion content of snow at Summit, Greenland: Constraints from three years of daily surface snow samples, Atmos. Environ., 41, 5007-5019). As suggested by the title, this paper reports on 3 years, through the winters, of daily snow sampling at Summit, including ion concentrations and fluxes, and specifically separates freshly fallen from surface snow aged days to weeks. It probably needs to be considered in relation to the summary of previous work at top of page 2, and contradicts the statement made in lines 4-5 on page 3. Also, a main point of Dibb et al., 2007 was to compare seasonality of ions in snow falling at Summit to the seasonality inferred from snowpit studies, so it would seem an additional valid source for the comparisons made at the end of section 3.1 in this manuscript (page 9, lines 5-15).

Response: We apologize for missing the Dibb et al. (2007) study in our last revision. This reference has been added to the discussion in section 3.1 and Table S5.

Dibb, J. E., Whitlow, S. I., and Arsenault, M.: Seasonal variations in the soluble ion content of snow at Summit. Greenland: Constraints from three years of daily surface snow samples, Atmos. Environ., 41, 5007-5019, doi:10.1016/j.atmosenv.2006.12.010, 2007.

**Response to Detailed Comments**

Referenced to Page/Line #(s) in the original manuscript:

**3/30**

Referee Comment: is unavoidable and natural within natural snowpack systems. → is unavoidable within natural snowpacks and still expected to smaller extent on the snow table.

Response: Line corrected in text. (revised manuscript page/line 3/32)

**3/30-32**

Referee Comment: It is actually not necessary to know the depth of a sampled surface stratigraphic layer to calculate fluxes, in fact the depth can, and often does, vary horizontally. All that is needed is the area sampled and the mass of snow in that area (e.g., g snow/cm^2). Given concentrations in mass or mole per g of melted snow, conc x g snow/cm^2 give the amount of chemical in the layer per cm^2. The harder challenge is to only sample from single stratigraphic layer, but that can be done with care.

Original Line: *Also, the collection of samples from a snow table eliminated the need to estimate the depth of fresh snow, a source of uncertainty for traditional surface snow sampling.*

Response: This line was meant to touch on the second challenge described by the referee: the difficulty in sampling from a single stratigraphic layer. The line has been revised to clarify this point as follows: *Also, the collection of samples from a snow table eliminated the difficulty in distinguishing the fresh stratigraphic snow layer from aged layers below, a source of uncertainty for traditional surface snow sampling.* (4/1-2)

**4/3-7**

Referee Comment: Possible underestimate of snowfall up to a factor of 10 seems too large to just pass over. Here in the text it is stated that large discrepancies between operator reports and a met station separated by 6 km justify ignoring the station date, referring to section 4.2 in the supplement for details. However, there are no details on the discrepancies in the supplement, just a rephrased version of the same statement.

Response: The discussion of meteorological data in the supplemental has been expanded. We agree with the referee that the difference in snow depth may indicate under-catch by the snow table; however, observed significant differences in weather conditions between the collection site and meteorological station during the campaign as well as operator notes from previous years make us weary to simply adopt the meteorological station data for this analysis. Some dates with recorded precipitation at the collection site showed none at the meteorological station, and vice-versa. To provide better confidence in the analysis we have included a revised version of Figure 3 in the supplemental (revised Figure S3) which uses meteorological station snow depths instead of those measured at the snow table. The results show a change in absolute value of the effective deposition velocities, though largely within the uncertainty shown in the original plot. Yet, interpretation of the data remains largely unchanged as the relative values between analytes are unaffected.

**4/30**

Referee Comment: Be more specific what you mean by underestimate of BC mass due to size cut-off. First point is that 0.02-50 fg is not directly a size (need to assume shape factor and density). Also not clear whether you are worried about truncation on the small end or the large end of the distribution (both could be possible). It also seems that the cited mass range is based on the SP2 detection principle, probably also need to think about any issues related to transmission efficiency of the nebulizer.

Original Line: *Observed BC mass distributions did not suggest significant underestimation of the total BC mass due to this size cut-off.*

Response: Details on the SP2 analysis are provided in the supplemental section S2.1.1. This section provides the density and shape assumptions and a description of the cut-off concern. As stated in the text, the mass outside the detectable range was found to be negligible, both above and below.

**7/3-6 and Table 1**

Referee Comment: It is a little confusing that the list in the text does not overlap very well with the elements in the table (e.g. Ba, Ti, K, Cd, Ca, Cr, and Na are in text but not table while V, Se, Sb, and Tl are in the table but not listed in the text). Could either make the list in text agree with this table, or expand the list to include all of the elements in supplemental tables S3 and S4 and make it clear you are referring to those tables and not Table 1.

Response:  We apologise for this oversight. The line in question has been corrected to discuss only those metals included in Table 1. The metals Se, Sb, and Tl have been left out of this list due to insufficient filtrate measurements above detection limit. Since Table 1 only described the insoluble portion of these metals there inclusion in the table is considered acceptable.

Corrected Line: *The metal measurements can be roughly classified into three categories: predominantly insoluble analytes Fe and Al (>50% insoluble over full campaign); variably soluble/insoluble analytes Co, V, As, Cu, Pb, Mn, K, and Mg; and predominantly soluble analytes Ca and Na (<50% insoluble) (in order from least to greatest average soluble fraction), excluding analytes with insufficient soluble or insoluble measurements above MDL.* (7/5-9)

**Table 1 Footnote**

Referee Comment: MSA, ACE, PRP, and FOR are abbreviations, so OK to define, but $C_2O_4^-$ and $NH_4^+$ are not. If you need to define these ion formulae, you probably need to define all of them. Better choice would seem to stop after the first 4.

Related comment, MSA usually signifies methylsulfonic acid, not the ion methylsulfonate that you correctly indicate was measured by IC. I think that $MS^-$ would be better shorthand to indicate methylsulfonate (here and everywhere else in the MS).

Response: The names for all abbreviations and all compounds have been provided in the text at their first appearance to provide clarity to the reader. The other abbreviations/compounds in Table 1 (BC, $NO_3^-$, and $SO_4^{2-}$) were named at their first appearance in section 1, but the chemical species listed above appear in the manuscript for the first time in Table 1. However, we agree that naming compounds is not strictly necessary and that Tables and the abbreviations therein should be stand-alone if possible. Thus, all abbreviations with Table 1 have been defined in the revised footnote and compound names have only been provided at their first appearance in the body of the text.

As per the referee's suggestion, the acronym MSA has been replaced with MS throughout the manuscript and supplemental.

**Section 3.2.2, 3.2.3, and 4**

Referee Comment: Need to be consistent and show the valences for all of the ions everytime they appear in the text. Right now only a few are shown. This is especially important since you also discuss results of elemental analyses for Na, K, Mg and Ca. Throughout these 3 sections the references are all to the ions, not elements.

Response: We agree that adding valences for all chemical species measured by IC would provide better consistency throughout the discussion. We had hoped to avoid readers misinterpreting the inclusion of valences as an indication of the expected form in the environment; however, if the exclusion of valence causes greater confusion then we are happy to add them back in. The exception is when chemical species are being discussed in general, not specifically as measurements (for example "...*Cl partitioned to the gas phase...*" or "*... Ca-rich mineral dust...*"). This exception may lead to a lack of consistency but we believe that incorrect inclusion of valence in such a discussion would lead to greater confusion. We have previously added the following line in hope to discourage readers from misinterpreting any included valences as an indication of the expected in-situ form: "*It should be noted that although IC measurements are provided as the measured ions throughout the discussion, these analytes may not necessarily exist in the dissociated ionic form in the environment.*"

**12/6**

Referee Comment: Not so clear that NH4^+ is always dominant compared to NH3. Maybe in the Arctic, and during cold season, but this reads like a global declaration.

Original Line: *First, the measured chemical species differ in terms of dominant phase: BC, $NH_4^+$, $SO_4^{2-}$, Na, K, Mg, $C_2O_4^{2-}$, and Ca are typically observed predominantly in the particle phase, while MSA, Br, Cl, and $NO_3^-$ and their associated precursors can have appreciable gas-phase portions (Barrie and Hoff, 1985).*

Response: This line has been revised to clarify that the predominant phases discussed are in reference to within the Arctic.

Revised Line: *First, the measured chemical species differ in terms of dominant phase: BC, $NH_4^+$ (ammonium), $SO_4^{2-}$, $Na^+$, $K^+$, $Mg^{2+}$, $C_2O_4^{2+}$ (oxalate), and $Ca^{2+}$ are typically observed predominantly in the particle phase within the Arctic, while MSA, $Br^-$, $Cl^-$, and $NO_3^-$ and their associated precursors can have appreciable gas-phase portions (Barrie and Hoff, 1985). (12/5-8)*

**12/8-9**

Referee Comment: As noted earlier, Ca^2+ is a striking exception to this statement, for no plausible reason. Do you really think this is real?

Response: See response to comment above.

**13/8**

Referee Comment: Seems "in-cloud scavenging" should be "below-cloud"

[revised manuscript text omitted]